# Pattern of tamoxifen-induced Tie2 deletion in endothelial cells in mature blood vessels using endo SCL-Cre-ERT transgenic mice

Peter J. Zwiers[1]☯*, Rianne M. Jongman[1,2,3]☯, Timara Kuiper[1], Jill Moser[1,2], Radu V. Stan[4], Joachim R. Göthert[5], Matijs van Meurs[1,2], Eliane R. Popa[1], Grietje Molema[1]

**1** Department of Pathology and Medical Biology, Medical Biology Section, University Medical Center Groningen, University of Groningen, Groningen, The Netherlands, **2** Department of Critical Care, University Medical Center Groningen, University of Groningen, Groningen, The Netherlands, **3** Department of Anesthesiology, University Medical Center Groningen, University of Groningen, Groningen, The Netherlands, **4** Geisel School of Medicine at Dartmouth, Department of Biochemistry and Cell Biology, One Medical Center Drive, Lebanon, NH, United States of America, **5** Department of Hematology and Stem Cell Transplantation, West German Cancer Center, University Hospital Essen, Essen, Germany

☯ These authors contributed equally to this work.
* p.j.zwiers@umcg.nl

**Data Availability Statement:** All relevant data are within the paper and its Supporting Information files. Additional large data sets are made available

## Abstract

Tyrosine-protein kinase receptor Tie2, also known as Tunica interna Endothelial cell Kinase or TEK plays a prominent role in endothelial responses to angiogenic and inflammatory stimuli. Here we generated a novel inducible *Tie2* knockout mouse model, which targets mature (micro)vascular endothelium, enabling the study of the organ-specific contribution of Tie2 to these responses. Mice with floxed *Tie2* exon 9 alleles (*Tie2^floxed/floxed*) were crossed with *end-SCL-Cre-ER^T* transgenic mice, generating offspring in which *Tie2* exon 9 is deleted in the endothelial compartment upon tamoxifen-induced activation of Cre-recombinase (*Tie2^ΔE9*). Successful deletion of *Tie2* exon 9 in kidney, lung, heart, aorta, and liver, was accompanied by a heterogeneous, organ-dependent reduction in Tie2 mRNA and protein expression. Microvascular compartment-specific reduction in Tie2 mRNA and protein occurred in arterioles of all studied organs, in renal glomeruli, and in lung capillaries. In kidney, lung, and heart, reduced Tie2 expression was accompanied by a reduction in *Tie1* mRNA expression. The heterogeneous, organ- and microvascular compartment-dependent knockout pattern of Tie2 in the *Tie2^floxed/floxed*;*end-SCL-Cre-ER^T* mouse model suggests that future studies using similar knockout strategies should include a meticulous analysis of the knockout extent of the gene of interest, prior to studying its role in pathological conditions, so that proper conclusions can be drawn.

## Introduction

Tyrosine-protein kinase receptor Tie2, also known as Tunica interna Endothelial cell Kinase, or TEK, is expressed by vascular endothelial cells, as well as other cell subsets of the hematopoietic lineage [1, 2]. Endothelial Tie2 plays a prominent role in blood vessel development,

via the Zenodo public repository, DOI 10.5281/zenodo.6563056.

**Funding:** The author(s) recieved no specific funding for this work.

**Competing interests:** The authors have declared that no competing interest exist.

vascular integrity, and endothelial responses to inflammation [3–6]. The current model of endothelial Tie2 engagement in signal transduction describes binding of non-endothelially produced agonist angiopoietin (Ang)-1 to Tie2 under quiescent conditions [7, 8]. Binding of Ang-1 induces autophosphorylation of Tie2, resulting in PI3-Akt signal transduction and blood vessel stabilization, ensuring vascular integrity [9–13]. In acute inflammation, such as prevails in, for example, sepsis, endothelially stored Ang-2 is rapidly released from Weibel-Palade bodies and competes with Ang-1 for binding to Tie2 [14–17]. As a result, Tie2 activation is aborted, and endothelial barrier function and vascular integrity are lost. Tie2 signaling is not only controlled by a balance between levels of Ang-1 and Ang-2, but also by heterodimerization of Tie2 with Tyrosine-protein kinase receptor Tie1, which stabilizes Tie2 autophosphorylation and downstream signaling [7].

Experimental models of endotoxemia and clinical sepsis are associated with microvascular barrier instability and leakage. These microvascular pathologies may be caused by the destabilizing effect of the Tie2-Ang2 interaction during acute inflammation, but also by the transient loss of Tie2 expression [14, 18–20]. The question arises how microvascular loss of Tie2 expression affects consecutive inflammatory events in sepsis patients, in whom vascular inflammatory responses are further complicated by secondary infections [21]. Given that this question cannot be directly addressed in clinical sepsis, mouse models combining conditional genetic deletion of *Tie2* with sepsis models are indispensable.

The aim of this study was to generate and characterize an inducible, endothelial-specific *Tie2* knockout mouse model, to allow future studies into the role of Tie2 in mature microvasculature [22] in e.g., organ failure in experimental and human sepsis. To achieve this aim, we crossed our previously reported *Tie2$^{floxed/floxed}$* [23] mice with *end-SCL-Cre-ER$^T$* mice [24]. Since in the latter the activation of Cre-recombinase fused to an estrogen receptor variant (Cre-ER$^T$) in the endothelial compartment is driven by tamoxifen, we thus obtained tamoxifen-inducible *Tie2$^{ΔE9}$* mice. After activation of Cre-recombinase by tamoxifen treatment, we explored the extent and location of *Tie2* knockout in kidney, lung, heart, liver, and aorta at whole organ level, and at the level of microvascular compartments. Furthermore, we investigated whether endothelial *Tie2* knockout affected expression levels of associated members of the Ang/Tie2 system, *i.e.* Tie1, Ang1, and Ang2. Our results demonstrate reproducible, heterogeneous, organ- and microvascular compartment-dependent reduction of Tie2 mRNA and protein levels, which calls for a thorough analysis of knockout patterns of genes of interest in such knockout models, before investigating their role in pathological conditions.

## Materials and methods

### Generation of *Tie2$^{ΔE9}$* knockout mice

To generate mice that allow tamoxifen-inducible deletion of *Tie2* exon 9 in the endothelial compartment, *Tie2$^{floxed/floxed}$* mice [23] were crossed with heterozygous *end-SCL-Cre-ER$^T$* transgenic mice [24]. Endothelial cell-specific knockout of *Tie-2* exon 9 (*Tie2$^{floxed/floxed}$*; *end-SCL-Cre-ER$^{T\ +/-}$*) was induced by intraperitoneal injection of tamoxifen (Sigma-Aldrich, Saint Louis, MO, USA) at a dose of 4 mg/0.1ml sterile corn oil (Sigma-Aldrich), three times a week, for a period of three weeks. Resulting knockout mice (n = 7; 2/7 female; 10–24 weeks old) are named *Tie2$^{ΔE9}$* in the rest of the manuscript. Body weights were assessed every 2–3 days during tamoxifen treatment. Twenty-eight days after the start of tamoxifen treatment, mice were sacrificed under inhalation anesthesia, organs were harvested, snap frozen on liquid nitrogen, and stored at -80˚C until further processing.

To obtain initial proof of deletion of *Tie2* exon 9 in *Tie2$^{ΔE9}$* mice (n = 3), genomic DNA was isolated from 10 μm kidney cryosections and PCR was done according to standard

protocols, using the primers 5'-GGGCTGCTACAATAGCTTTGG-3' and 5'-GTTATGTCCA GTGTCAATCAC-3'. Deletion of *Tie2* exon 9 in kidney, lung, heart, aorta, and liver of *Tie2$^{ΔE9}$* knockout mice and control mice that lacked Cre expression (*Tie2$^{floxed/floxed}$*; *end-SCL-Cre-ER$^{T-/-}$*, named *Tie2$^{fl/fl/Cre-}$* in the rest of the manuscript; n = 8; 3/8 female; 10–24 weeks old) was investigated using the same PCR method.

## RNA isolation and quantification of gene expression by RT-qPCR

Total RNA was isolated from organs using the RNeasy® Plus Mini Kit and total RNA from laser-microdissected material (see below) was isolated using the RNeasy® Plus Micro Kit, according to the manufacturer's protocols (QIAGEN, Venlo, The Netherlands). RNA concentration and purity were measured on a ND1000 UV-VIS system (NanoDrop Technologies, Rockland, DE, USA). RNA integrity was determined by agarose gel electrophoresis. cDNA was synthesized using random hexamer primers (Promega, Leiden, the Netherlands) and the SuperScript-III RT Kit, according to the manufacturer's protocol (Life Technologies Europe B. V, Bleiswijk, The Netherlands). Duplicate qPCRs were performed for each sample on the ViiA7 Real-Time PCR system (Thermo Fisher Scientific, Bleiswijk, The Netherlands) using Assay-on-Demand primer/probe sets (Thermo Fisher Scientific; Table 1). To determine the possible formation of truncated *Tie2* mRNA upon exon 9 deletion, we used 3 different primer/ probe sets that hybridize at cDNA sites before (at the boundary between exon 7 and exon 8, hereafter referred to as 'exon 7/8'), at (exon 8/9), or after exon 9 (exon 15/16) (see Table 1).

Cycle threshold values ($C_t$) of duplicate reactions were averaged. Gene expression of *Cdh5* (also known as *VE-cadherin)* was normalized to the expression of the reference gene *Gapdh*. Expression of the endothelium-associated genes *Tek* (here after called *Tie2), Tie1, Angpt1* and *Angpt2* (hereafter called *Ang1* respectively *Ang2*), and *Pecam1 (*also known as *CD31*) was normalized to the expression of *VE-cadherin*. Relative mRNA expression was calculated using the formula $2^{-ΔCt}$ [25].

## Isolation of renal microvascular compartments by laser microdissection

To assess the deletion of *Tie2* exon 9 and *Tie2* mRNA levels in arterioles and glomeruli of the kidney by RT-qPCR, these microvascular compartments were isolated by laser microdissection (LMD), as described previously [26].

## Tie2 protein quantification by ELISA

Tie2 ELISA was performed as described previously [23]. Briefly, cryosections of organs were used to prepare tissue homogenates in RIPA buffer containing protease inhibitor, phosphatase inhibitor and activated $Na_3VO_4$. Total protein concentration was determined by *DC*™ Protein

**Table 1. RT-qPCR primers.**

| Gene | Assay ID | Encoded protein |
|------|----------|-----------------|
| *Gapdh* | Mm99999915_g1 | Glyceraldehyde-3-phosphate dehydrogenase (Gapdh) |
| *Tek* | Mm00443242_m1 | Tyrosine kinase receptor (Tie2), CD202 (exon 7/8) |
| *Tek* | Mm00443243_m1 | Tyrosine kinase receptor (Tie2), CD202 (exon 8/9) |
| *Tek* | Mm01256894_m1 | Tyrosine kinase receptor (Tie2), CD202 (exon 15/16) |
| *Tie1* | Mm00441786_m1 | Tyrosine kinase receptor 1 |
| *Angpt1* | Mm00456503_m1 | Angiopoietin 1 (Ang1) |
| *Angpt2* | Mm00545822_m1 | Angiopoietin 2 (Ang2) |
| *Pecam1* | Mm00476702_m1 | Platelet and Endothelial Cell Adhesion Molecule 1 (CD31) |
| *Cdh5* | Mm00486938_m1 | Cadherin 5 (VE-cadherin) |

Assay Kit II (#5000112; Bio-Rad Laboratories B.V., Veenendaal, The Netherlands). Tie2 protein expression was quantified using Mouse Tie2 Quantikine ELISA Kit (cat# MTE200, sensitivity 40.2 pg/ml and assay range 125.0–8,000 pg/ml), according to manufacturer's instruction (R&D Systems, Minneapolis, MN, USA). Amounts of Tie2 protein were normalized for the total protein input of tissue homogenate and expressed as pg/μg of total protein. Raw data files are available in supplementary data files (S1 Dataset).

## Histological localization of Tie2 protein by immunohistochemistry

To determine the histological localization of Tie2 protein in distinct microvascular compartments, immunohistochemical staining of Tie2 was performed as previously described [18]. Briefly, 4 μm acetone-fixed cryosections were incubated with primary monoclonal rat-anti-Tie2 antibody (Tek4, IgG1 isotype; eBioscience, Thermo Fisher Scientific, San Diego, CA, USA), followed by rabbit-anti-rat IgG1 antibody (Vector Laboratories, Burlingame, CA, USA), and anti-rabbit, horseradish peroxidase-labeled polymer (Dako, Heverlee, Belgium). Peroxidase activity was detected with 3-Amino-9-ethylcarbazole (Sigma-Aldrich). Sections were counterstained with Mayer's hematoxylin (Merck, Darmstadt, Germany). Isotype control staining with rat IgG1 (Antigenix America, New York, USA) was consistently negative. All sections of a given organ (n = 15 mice) were stained in one experiment to avoid inter-experiment staining variability.

## Morphometric quantification of Tie2 staining in microvascular compartments

Stained sections were scanned with a NanoZoomer® 2.0 HT (Hamamatsu Photonics, Almere, The Netherlands). Raw NPDI files are available via repository DOI 10.5281/zenodo.6563056. Immunohistochemical staining was quantified as previously described, using Aperio Imagescope software v12.2 (Leica Biosystems Imaging, Vista, CA, USA) [23]. Briefly, regions of interest were drawn around the perimeter of entire tissue sections or specific vascular compartments (*e.g.* glomeruli), excluding occasional artifacts (tissue breaks or folds). To quantify Tie2 staining in capillaries, random regions of interest were drawn in tissues, excluding arterioles, venules, and, in kidneys, glomeruli. After automated counting of pixels, the percentage of positive pixels per total tissue section or vascular bed type was calculated.

## Statistical analysis

Normality of distribution was tested using the Shapiro-Wilk normality test on *Tie2* mRNA data obtained from whole organ extracts and LMD isolates from the kidney. The results showed that values are normally distributed (data not shown). Statistically significant differences in mRNA and protein levels between *Tie2*$^{fl/fl/Cre-}$ control mice and *Tie2*$^{\Delta E9}$ knockout mice were calculated by a two-tailed unpaired *t*-Test. Statistically significant differences in mRNA levels between organs were calculated by one-way ANOVA with Sidak correction for multiple comparisons. Correlations were calculated using the Pearson correlation test. Statistics were performed using GraphPad Prism 9.2.0 (GraphPad Prism Software Inc. La Jolla, CA, USA). Differences were considered statistically significant when $p<0.05$. All GraphPad PZFX-files are available as supplementary data (S2–S8 Dataset).

## Study approval

All experimental procedures were approved and performed in accordance with the University of Groningen Animal Ethical Committee (DEC) for the use of experimental animals and the national animal testing act (WOD) and European guidelines for animal care and use.

## Results

### Generation and validation of the inducible endothelial-specific Tie2 knockout mouse model

We previously generated a $Tie2^{+/-}$ heterozygous knockout mouse by crossing floxed Tie2 exon 9 ($Tie2^{floxed/floxed}$) mice with *Hprt-Cre* mice. Upon intercross of $Tie2^{+/-}$ mice, no $Tie2^{-/-}$ double knockout mice were born, confirming that the deletion of Tie2 exon 9 was functional, i.e. it resulted in embryonic lethality. Furthermore, heterozygous $Tie2^{+/-}$ mice had 50% less Tie2 mRNA and protein compared to littermate controls [23].

To generate a transgenic mouse that allows conditional knockout of *Tie2* in mature microvascular endothelium, we crossed $Tie2^{floxed/floxed}$ mice with *end-SCL-Cre-ER$^T$* mice (Fig 1A) [24]. Upon tamoxifen-induced activation of Cre-recombinase, exon 9 of *Tie2* ($Tie2^{\Delta E9}$ knockout mice) was successfully deleted in kidney (Fig 1B and S1 Fig), lung, heart, aorta, and, to a lesser extent, in liver (S1 Fig). In littermate control mice lacking Cre-recombinase ($Tie2^{fl/fl/Cre-}$ control mice), the *Tie2* locus remained intact in all studied organs (Fig 1B and S1 Fig). Tamoxifen treatment and deletion of *Tie2* exon 9 did not affect body weight (S2 Fig) and did not result in altered behavior.

### Tamoxifen-induced activation of Cre-recombinase results in organ-specific reduction of Tie2 expression

Our previous proof of functional Tie2 deletion in $Tie2^{+/-}$ heterozygous knockout mice [23] led us to expect reduced Tie2 mRNA and protein expression after tamoxifen-induced deletion of *Tie2* exon 9 in $Tie2^{\Delta E9}$ knockout mice. Since our inducible knockout mouse targets the endothelium, and since vascular density differs per organ, we used the pan-endothelial marker gene *VE-cadherin* as a reference gene for the quantification of *Tie2* mRNA levels in studied organs. We confirmed the validity of *VE-cadherin* as reference gene by showing that the mRNA expression of *VE-cadherin* was not altered in $Tie2^{\Delta E9}$ knockout mice compared to $Tie2^{fl/fl/Cre-}$ control mice (S3A Fig). In addition, mRNA levels of *CD31* (*Pecam1*), another pan-endothelial molecule, were not affected in $Tie2^{\Delta E9}$ knockout mice, with the exception of the kidney, where reduced expression of *CD31* was minor, albeit statistically significant (S3B Fig).

In $Tie2^{fl/fl/Cre-}$ control mice, *Tie2* mRNA expression was highest in kidney (Fig 2A). Tie2 expression in the other four organs was approximately 4-fold lower than in the kidney and did not differ significantly between these organs (S1 Table shows corresponding *p*-values). In $Tie2^{\Delta E9}$ knockout mice, *Tie2* mRNA expression was significantly reduced in kidney (25%; mean $Tie2^{\Delta E9}$ = 0.59 (SD 0.08); mean $Tie2^{fl/fl/Cre-}$ = 0.79 (SD 0.12); $p = 0.0026$), lung (55%; mean $Tie2^{\Delta E9}$ = 0.12 (SD 0.05); mean $Tie2^{fl/fl/Cre-}$ = 0.27 (SD 0.06); $p = 0.0002$), heart (35%; mean $Tie2^{\Delta E9}$ = 0.15 (SD 0.03); mean $Tie2^{fl/fl/Cre-}$ = 0.23 (SD 0.03); $p < 0.0001$), and aorta (38%; mean $Tie2^{\Delta E9}$ = 0.13 (SD 0.07); mean $Tie2^{fl/fl/Cre-}$ = 0.21 (SD 0.04); $p = 0.013$), but not in liver (mean $Tie2^{\Delta E9}$ = 0.17 (SD 0.04); mean $Tie2^{fl/fl/Cre-}$ = 0.18 (SD 0.05); $p = 0.54$) (Fig 2A). This organ-specific reduction of *Tie2* mRNA was paralleled by a reduction in Tie2 protein in lung (66%; mean $Tie2^{\Delta E9}$ = 2.45 (SD 1.10); mean $Tie2^{fl/fl/Cre-}$ = 7.21 (SD 3.01); $p = 0.0017$), heart (37%; mean $Tie2^{\Delta E9}$ = 0.31 (SD 0.05); mean $Tie2^{fl/fl/Cre-}$ = 0.50 (SD 0.06); $p < 0.0001$), and aorta (51%; mean $Tie2^{\Delta E9}$ = 0.36 (SD 0.18); mean $Tie2^{fl/fl/Cre-}$ = 0.73 (SD 0.22); $p = 0.0033$), as assessed by ELISA on whole organ homogenates (Fig 2B).

To confirm that reduced levels of *Tie2* originate from residual full length *Tie2* and is not a result of the formation of truncated *Tie2*, we performed RT-qPCR analysis on lung RNA isolates from $Tie2^{fl/fl/Cre-}$ control mice and $Tie2^{\Delta E9}$ knockout mice. For this analysis we used three primer/probe sets that bind to different intron-overspanning regions within the *Tie2* cDNA.

**A**

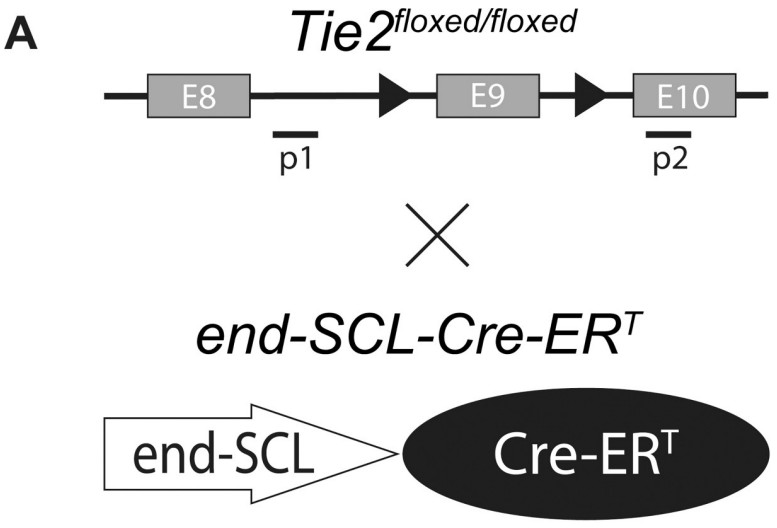

**B**

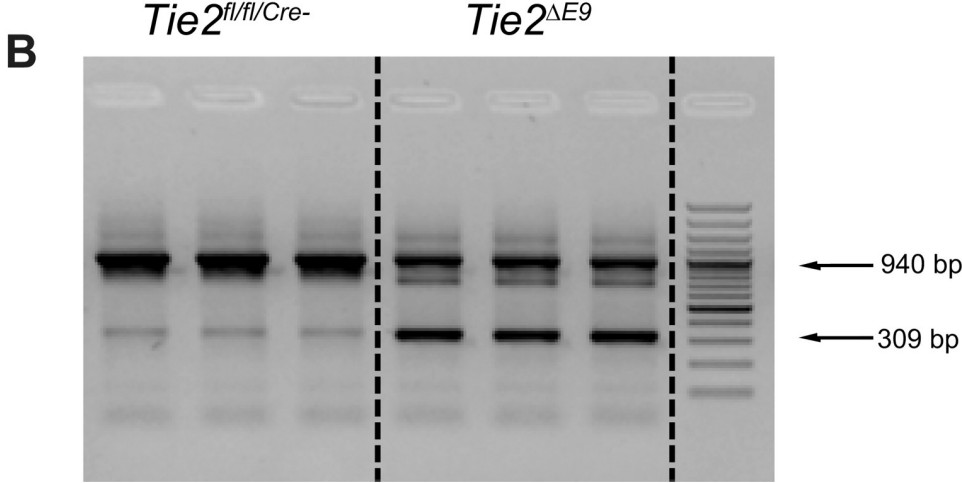

**Fig 1. Generation of the endothelial-specific, inducible *Tie2* knockout mouse.** The generation of mice with floxed *Tie2* exon 9 (*Tie2^floxed/floxed^*) was described previously [23]. **(A)** *Tie2^floxed/floxed^* mice were crossed with *end-SCL-Cre-ER^T^* transgenic mice, which express Cre-recombinase in the endothelial compartment [24]. P1 and p2 refer to primers used to detect the presence/absence of *Tie2* exon 9. **(B)** After tamoxifen-induced activation of Cre-recombinase, the presence (a 940 basepair (bp) PCR product) or absence (a 309 bp PCR product) of *Tie2* exon 9 was confirmed by genomic PCR in kidney of *Tie2^fl/fl/Cre-^* control mice (n = 3) and *Tie2^ΔE9^* knockout mice (n = 3). Lanes show PCR products for individual mice.

While they hybridized to the Tie2 cDNA to different extents, the percentage of *Tie2* downregulation in the *Tie2^ΔE9^* knockout mice versus controls measured by each of these primer/probe sets was approximately 50% and comparable to one another (S4 Fig). We also performed Western Blot analysis on protein isolates from the lung, and did not detect smaller size, truncated Tie2 protein in *Tie2^ΔE9^* knockout mice compared to controls (data not shown). Based on these results we concluded that the remaining, detectable Tie2 mRNA and protein in our mouse model represent residual full length Tie2.

In conclusion, deletion of *Tie2* exon 9 in our *Tie2^floxed/floxed^;end-SCL-Cre-ER^T^* knockout mouse model resulted in significant, organ-specific reduction of Tie2 mRNA and protein expression.

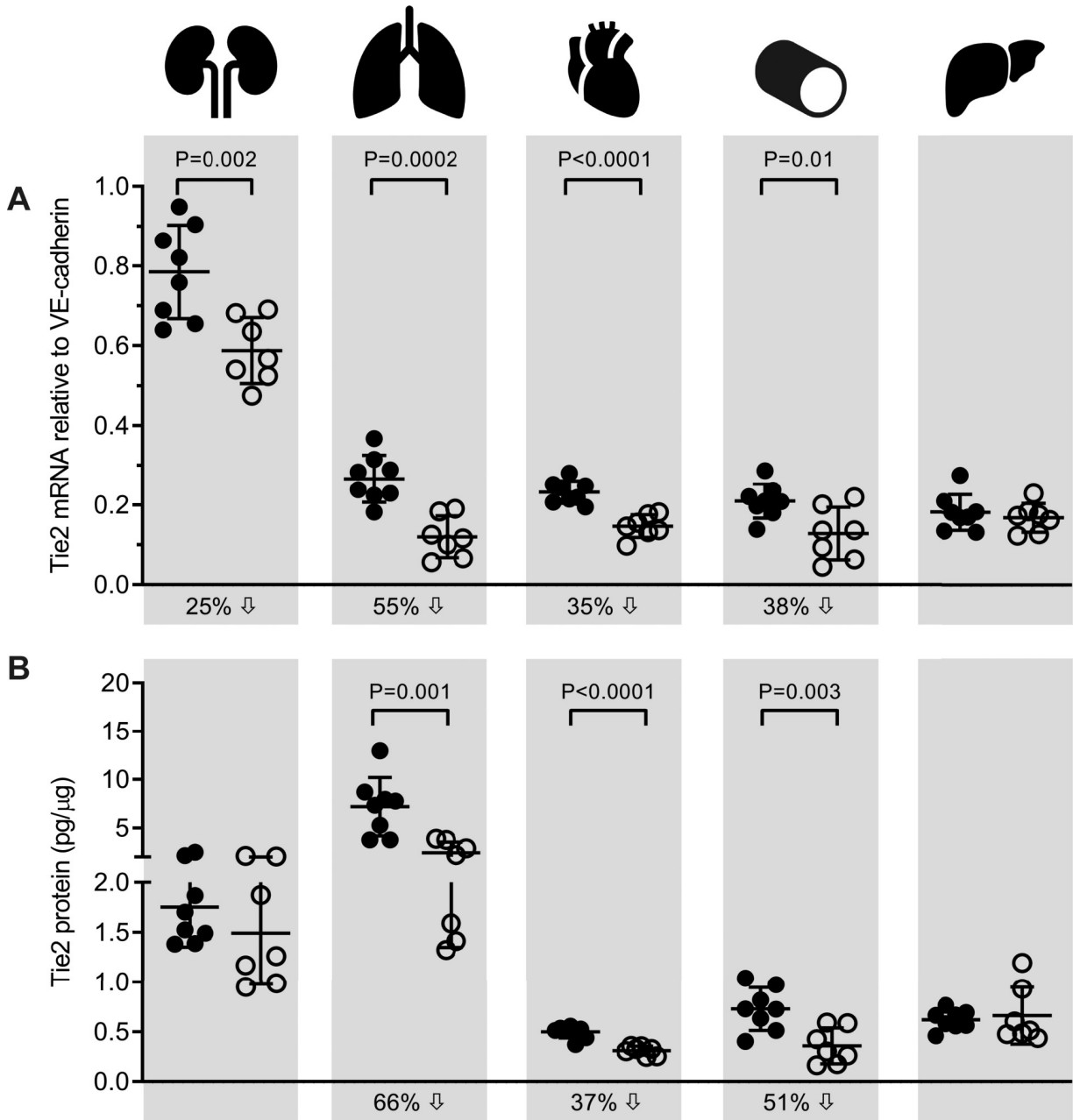

**Fig 2. Tie2 mRNA and protein expression in control and *Tie2* knockout mice.** Tie2 expression levels were determined in organs of *Tie2*<sup>fl/fl/</sup> *Cre-* control mice and *Tie2*<sup>ΔE9</sup> knockout mice after tamoxifen treatment by (**A**) RT-qPCR, and (**B**) ELISA. Graphs show individual values and means (black lines) ± SD (error bars). Percent decrease (downward arrow) of Tie2 mRNA or protein in *Tie2*<sup>ΔE9</sup> knockout mice compared to *Tie2*<sup>fl/fl/Cre-</sup> controls is indicated under the X-axes. Closed circles: *Tie2*<sup>fl/fl/Cre-</sup> mice (n = 8); open circles: *Tie2*<sup>ΔE9</sup> mice (n = 7).

### Reduction of Tie2 protein expression in *Tie2*<sup>ΔE9</sup> knockout mice occurs in specific microvascular compartments

Since in organs of *Tie2*<sup>ΔE9</sup> knockout mice Tie2 protein expression was reduced to different extents (Fig 2B), we next asked in which microvascular compartments of these organs this reduction occurred. Immunohistochemical staining showed that Tie2 was expressed robustly in arterioles and venules, and variably in capillaries of *Tie2*<sup>fl/fl/Cre-</sup> control organs (Fig 3A).

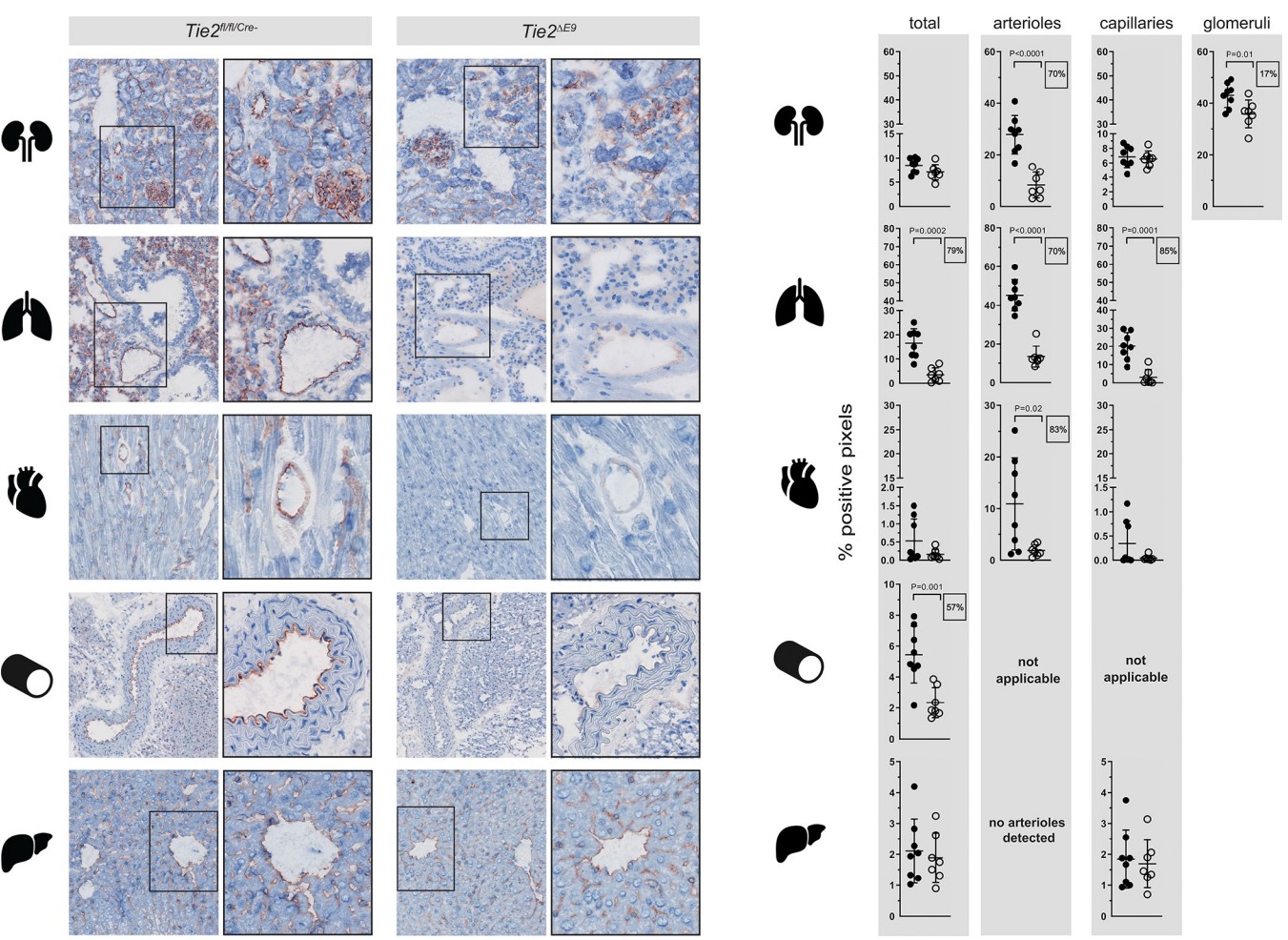

**Fig 3. Expression patterns of Tie2 protein in organs of control and *Tie2* knockout mice.** Tie2 protein was detected by immunohistochemistry in organs of *Tie2^{fl/fl/Cre-}* control mice and *Tie2^{ΔE9}* knockout mice after tamoxifen treatment. (**A**) Photomicrographs of Tie2 staining, taken at 200x optical magnification, on the left. For each genotype, right-column photomicrographs show digital magnifications of regions of interest denoted by boxes in the left column. (**B**) Positive pixels representing Tie2 staining were quantified by morphometry of whole tissue sections (total) and separate microvascular compartments within organs (arterioles and capillaries, glomeruli in the kidney). Graphs show individual values and means (black lines) ± SD (error bars). Closed circles: *Tie2^{fl/fl/Cre-}* mice (n = 8); open circles: *Tie2^{ΔE9}* mice (n = 7). Boxed numbers show percent reduction of Tie2 expression in specific vascular beds of *Tie2^{ΔE9}* mice compared to *Tie2^{fl/fl/Cre-}* mice.

Microscopic inspection of organs of *Tie2^{ΔE9}* knockout mice revealed a strong reduction in Tie2 protein in kidney, lung, heart, and aorta, and unaltered expression in liver (Fig 3A). When Tie2 protein staining was quantified by morphometry in whole organ sections of *Tie2^{ΔE9}* knockout and *Tie2^{fl/fl/Cre-}* control mice, a statistically significant Tie2 protein reduction was identified in lung (79%; mean *Tie2^{ΔE9}* = 3.50 (SD 2.83); mean *Tie2^{fl/fl/Cre-}* = 16.52 (SD 5.96); *p* = 0.0002) and aorta (57%; mean *Tie2^{ΔE9}* = 2.34 (SD 0.98); mean *Tie2^{fl/fl/Cre-}* = 5.45 (SD 1.83); *p* = 0.0015) (Fig 3B, 'total').

We next refined quantification of Tie2 protein by selectively focusing on microvascular compartments (Fig 3B). This approach uncovered a significant reduction in Tie2 protein expression in arterioles (70%; mean *Tie2^{ΔE9}* = 8.27 (SD 4.94); mean *Tie2^{fl/fl/Cre-}* = 27.81 (SD 7.46); *p*<0.0001) and glomeruli (17%; mean *Tie2^{ΔE9}* = 35.84 (SD 5.36); mean *Tie2^{fl/fl/Cre-}* = 43.04 (SD 4.75); *p* = 0.0163) of the kidney, which had been concealed by whole-tissue quantification. Moreover, a significant reduction in Tie2 protein was detected in arterioles

of the heart (83%; mean $Tie2^{\Delta E9}$ = 1.86 (SD 1.08); mean $Tie2^{fl/fl/Cre-}$ = 10.91 (SD 8.93); $p$ = 0.02), which, similarly, had been obscured by whole-tissue quantification. Vascular compartment-specific quantification in lung revealed a significant reduction of Tie2 protein in both arterioles (70%; mean $Tie2^{\Delta E9}$ = 13.51 (SD 5.41); mean $Tie2^{fl/fl/Cre-}$ = 45.24 (SD 8.05); $p$<0.0001) and capillaries (85%; mean $Tie2^{\Delta E9}$ = 3.01 (SD 4.22); mean $Tie2^{fl/fl/Cre-}$ = 20.18 (SD 7.45); $p$ = 0.0001) (Fig 3B).

In summary, we found a consistent, significant reduction of Tie2 protein in arterioles across organs, and a selective Tie2 reduction in capillaries of the lung and in renal glomeruli of $Tie2$-$^{floxed/floxed}$;$end$-$SCL$-$Cre$-$ER^T$ mice. This heterogeneity in endothelial $Tie2^{\Delta E9}$ knockout across mature microvascular compartments of various organs was concealed when analyzing the tissue at whole organ level.

## $Tie2^{\Delta E9}$ knockout selectively affects $Tie1$ mRNA expression

Signaling via Tie2 is dependent on its heterodimerization with the orphan receptor Tie1, and on the binding of the ligands Ang1 or Ang2 [7, 17, 27]. The significant reduction of Tie2 expression in several organs of $Tie2^{\Delta E9}$ knockout mice raised the question whether an adaptive response in expression of $Tie1$, $Ang1$, and $Ang2$ in these organs took place.

In $Tie2^{fl/fl/Cre-}$ control mice, mRNA levels of $Tie1$, $Ang1$, and $Ang2$ in kidney were significantly higher than those in the other organs (Fig 4; S1 Table shows corresponding $p$-values). In $Tie2^{\Delta E9}$ knockout mice, $Tie1$ mRNA expression was significantly reduced in kidney (19%; mean $Tie2^{\Delta E9}$ = 0.40 (SD 0.07); mean $Tie2^{fl/fl/Cre-}$ = 0.49 (SD 0.08); $p$ = 0.0303), lung (23%; mean $Tie2^{\Delta E9}$ = 0.16 (SD 0.01); mean $Tie2^{fl/fl/Cre-}$ = 0.21 (SD 0.03); $p$ = 0.0012) and heart (12%; mean $Tie2^{\Delta E9}$ = 0.35 (SD 0.04); mean $Tie2^{fl/fl/Cre-}$ = 0.40 (SD 0.03); $p$ = 0.0155) (Fig 4A), *i.e.* in organs that also exhibited a significant reduction in $Tie2$ mRNA levels. In contrast, the significant $Tie2$ mRNA reduction in aorta of $Tie2^{\Delta E9}$ knockout mice was not accompanied by a reduction in $Tie1$ mRNA expression. $Tie1$ mRNA expression in liver did not differ between genotypes.

Endothelial $Tie2^{\Delta E9}$ knockout did not affect $Ang1$- or $Ang2$ mRNA expression in most organs (Fig 4B and 4C), with the exception of the heart, where $Ang2$ mRNA expression was significantly increased (17%; mean $Tie2^{\Delta E9}$ = 0.0066 (SD 0.0007); mean $Tie2^{fl/fl/Cre-}$ = 0.0056 (SD 0.0008); $p$ = 0.0252) in $Tie2^{\Delta E9}$ knockout mice (Fig 4C). Cumulatively, these data show that reduction of Tie2 in kidney, lung, and heart in the $Tie2^{floxed/floxed}$;$end$-$SCL$-$Cre$-$ER^T$ mouse model, was accompanied by a reduction in expression of $Tie1$, but did not have a major effect on $Ang1$ and $Ang2$ expression.

## $Tie2^{\Delta E9}$ knockout differentially affects members of the Ang/Tie system in kidney microvascular compartments

The members of the Ang/Tie system, *i.c.* Tie1, Tie2, Ang1 and Ang2, play concerted roles in microvascular homeostasis [11–13]. Since we showed a reduction in Tie2 protein levels in arterioles and glomeruli of $Tie2^{\Delta E9}$ knockout mice (Fig 3B), we next asked whether this reduction affected the expression of other members of the Ang/Tie system in these specific microvascular compartments of the kidney. To answer this question, we isolated glomeruli and arterioles by laser microdissection and confirmed by genomic PCR that $Tie2$ exon 9 was deleted in both microvascular compartments (Fig 5A). Deletion of $Tie2$ exon 9 was accompanied by significantly reduced $Tie2$ mRNA levels in glomeruli (mean $Tie2^{\Delta E9}$ = 0.80 (SD 0.14); mean $Tie2^{fl/fl/Cre-}$ = 1.01 (SD 0.13); $p$ = 0.009). In arterioles from control mice high variation in Tie2 expression was observed, with no statistically significant differences between the two groups (mean $Tie2^{\Delta E9}$ = 0.67 (SD 0.19); mean $Tie2^{fl/fl/Cre-}$ = 0.87 (SD 0.36); $p$ = 0.21) (Fig 5B), contrasting

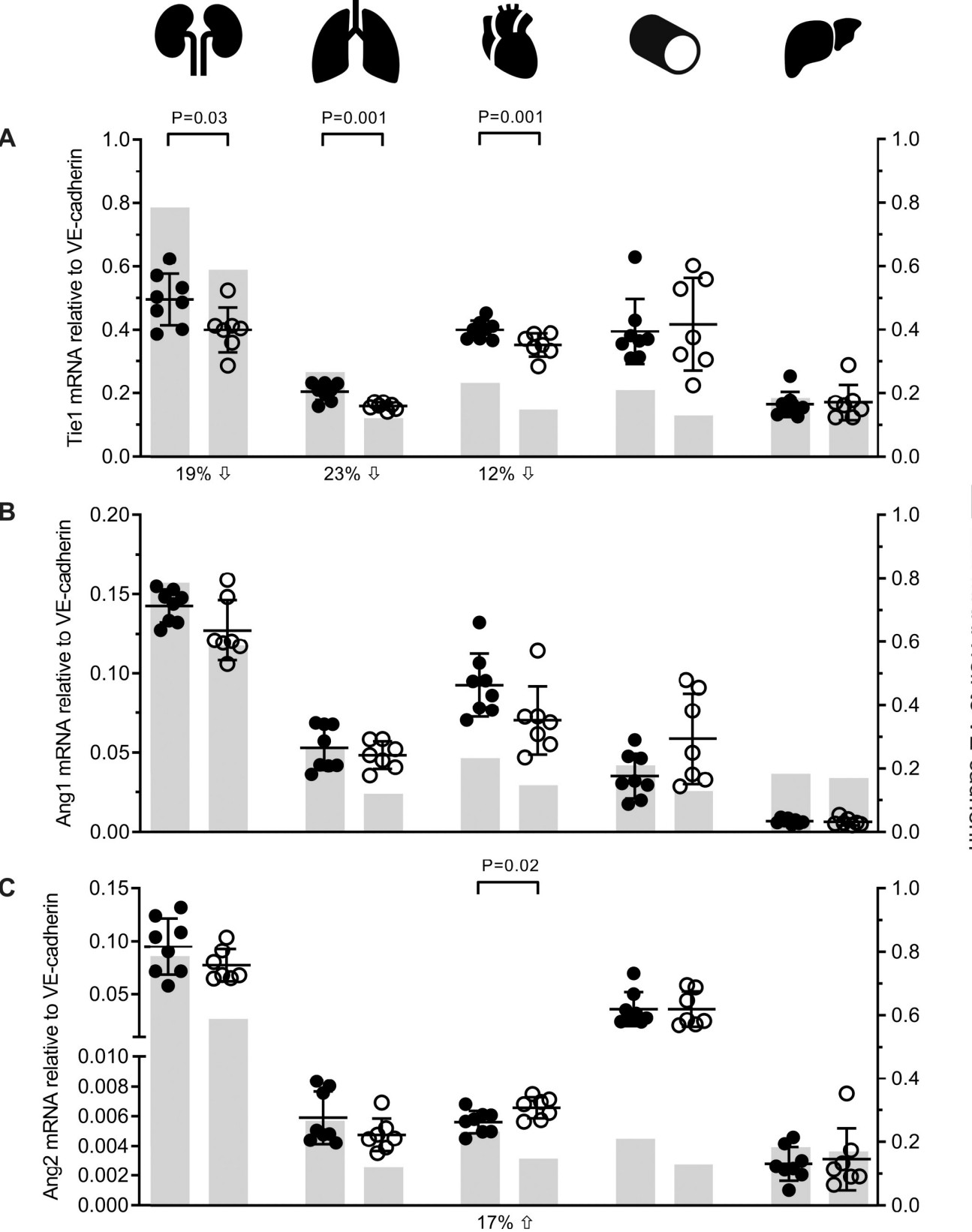

**Fig 4. Effect of Tie2 knockout on the expression of members of the Ang/Tie pathway in organs of *Tie2* knockout mice.** Expression of members of the Angiopoietin/Tie system was determined by RT-qPCR. (**A**) *Tie1* expression. Percent decrease (downward arrow) of *Tie1* mRNA in *Tie2*$^{ΔE9}$ knockout mice compared to *Tie2*$^{fl/fl/Cre-}$ control mice are indicated under the X-axis. (**B**) *Ang1* expression, and (**C**) *Ang2* expression. Percent increase (upward arrow) of *Ang2* mRNA in *Tie2*$^{ΔE9}$ knockout mice compared to *Tie2*$^{fl/fl/Cre-}$ control mice is indicated under the X-axis. Scatterplots show individual mRNA values and means (black lines) ± SD (error bars) of *Tie1/Ang1/Ang2* (left Y-axis). Grey bars serve as easy reference to mean values of *Tie2* expression (right Y-axis) in the respective organs as reported in Fig 2A. Closed circles: *Tie2*$^{fl/fl/Cre-}$ mice (n = 8); open circles: *Tie2*$^{ΔE9}$ mice (n = 7).

significantly lower Tie2 protein expression in the *Tie2*$^{ΔE9}$ knockout mice. *Tie1* mRNA expression was significantly increased in arterioles (mean *Tie2*$^{ΔE9}$ = 1.38 (SD 0.26); mean *Tie2*$^{fl/fl/Cre-}$ = 0.86 (SD 0.35); *p* = 0.007), but not in glomeruli (mean *Tie2*$^{ΔE9}$ = 0.40 (SD 0.03); mean *Tie2*$^{fl/fl/Cre-}$ = 0.47 (SD 0.08); *p* = 0.06) of *Tie2*$^{ΔE9}$ knockout mice (Fig 5C). Analysis of gene expression in kidney microvascular compartments also revealed a significant, two-fold increase in *Ang1* expression in arterioles of *Tie2*$^{ΔE9}$ knockout mice (mean *Tie2*$^{ΔE9}$ = 0.107 (SD 0.045); mean *Tie2*$^{fl/fl/Cre-}$ = 0.059 (SD 0.035); *p* = 0.036), but not in glomeruli (mean *Tie2*$^{ΔE9}$ = 0.022 (SD 0.007); mean *Tie2*$^{fl/fl/Cre-}$ = 0.023 (SD 0.008); *p* = 0.70) (Fig 5D), while *Ang2* expression was unchanged in arterioles (mean *Tie2*$^{ΔE9}$ = 0.016 (SD 0.012); mean *Tie2*$^{fl/fl/Cre-}$ = 0.017 (SD 0.011); *p* = 0.80) and glomeruli (mean *Tie2*$^{ΔE9}$ = 0.054 (SD 0.004), mean *Tie2*$^{fl/fl/Cre-}$ = 0.056 (SD 0.016); *p* = 0.78) (Fig 5E).

Since Tie2 functions in conjunction with Tie1, an interesting observation was the correlation between *Tie2* and *Tie1* mRNA levels (r = 0.84, *p* = 0.009) in arterioles, and the complete absence thereof in glomeruli of *Tie2*$^{fl/fl/Cre-}$ control mice (Fig 5F). The correlation between *Tie2* and *Tie1* expression in arterioles was even stronger in *Tie2*$^{ΔE9}$ knockout mice (r = 0.96, *p*<0.001; Fig 5G).

In conclusion, Tie2 knockout in mature arteriolar and glomerular endothelial cells of the kidney of *Tie2*$^{floxed/floxed}$;*end-SCL-Cre-ER*$^{T}$ mice was accompanied by changes in expression of *Tie1* and *Ang1* in a microvascular compartment-specific manner.

## Discussion

We generated a novel, endothelial-specific, inducible *Tie2* knockout mouse that enables future studies on how the absence/low expression of Tie2 in mature microvasculature affects endothelial and microvascular responses to pathophysiological conditions. Knockout was achieved by deletion of exon 9 of *Tie2* in the endothelial compartment. For this purpose, we used the *end-SCL-Cre-ER*$^{T}$ mouse model, in which tamoxifen-inducible Cre ER$^{T}$ recombinase expression was driven by the endothelial enhancer of the Stem Cell Leukemia locus. In this mouse model, using LacZ as reporter, Cre recombinase was shown to be active in the microvasculature of several, though not all, organs [24]. We mapped the patterns and extent of Tie2 expression loss in the microvasculature of kidney, lung, heart, liver, and in the aorta of *Tie2*$^{fl/fl/Cre-}$ control mice and *Tie2*$^{ΔE9}$ knockout mice, and investigated the consequences for the expression of *Tie1*, *Ang1*, and *Ang2*. Between 25% and 55% *Tie2* mRNA reduction in kidney, lung, heart, and aorta, was accompanied by a similar reduction of Tie2 protein in lung, heart, and aorta. In the microvascular compartments of these organs, more than 70% reduction in Tie2 protein expression occurred in arterioles of kidney, lung, and heart, and in lung capillaries. These changes were accompanied by 12–23% reduction in *Tie1* mRNA expression in kidney, lung, and heart. Interestingly, in aortic endothelium of knockout mice, loss of Tie2 protein did not affect *Tie1* mRNA expression. Zooming in further on microvascular segments of the kidney, reduction in Tie2 in arterioles was associated with an increase in *Tie1* and *Ang1* expression, an effect that was absent in glomeruli. Thus, our mouse model reveals a strikingly heterogeneous pattern of endothelial Tie2 knockout in mature microvasculature, displaying both inter- and

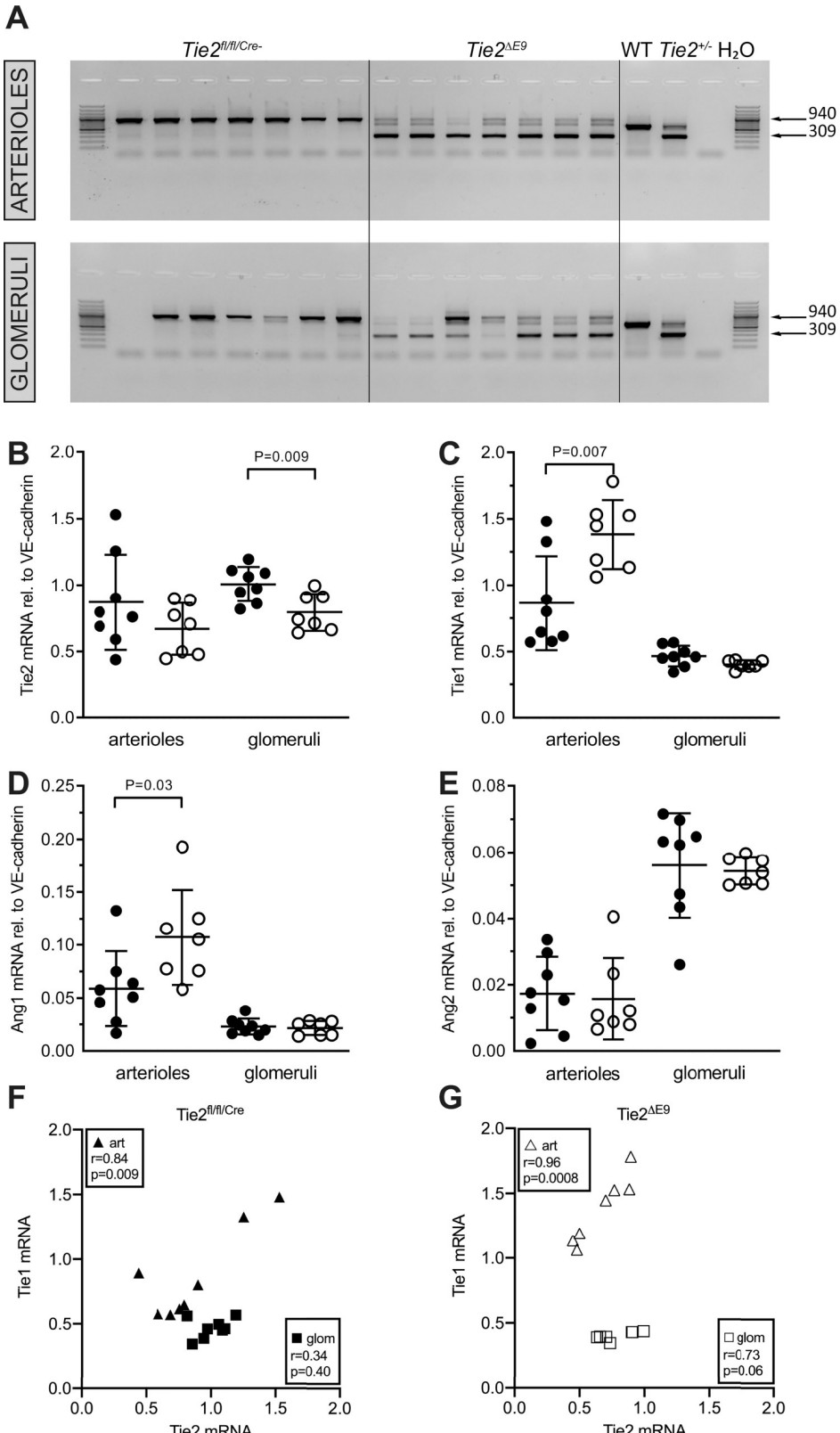

**Fig 5. Effect of Tie2 knockout on the expression of members of the Angiopoietin/Tie pathway in kidney microvascular compartments.** Arterioles and glomeruli were laser microdissected from kidney cryosections of *Tie2*<sup>fl/</sup>

*fl/Cre-* control mice and *Tie2^ΔE9^* knockout mice. (**A**) Deletion of *Tie2* exon 9 by tamoxifen-induced Cre-recombinase was confirmed by genomic PCR. A 309 basepair (bp) PCR product indicates deletion of exon 9, a 940 bp product the presence of exon 9. Lanes represent PCR products for individual mice. (**B-D**) Arteriolar and glomerular mRNA levels of *Tie2*, *Tie1*, *Ang1*, and *Ang2*, respectively. Graphs show individual values and means (black lines) ± SD (error bars). (**F, G**) Correlation plots for arteriolar and glomerular mRNA levels of *Tie2* and *Tie1* of (**F**) *Tie2^fl/fl/Cre-^* control mice, and (**G**) *Tie2^ΔE9^* knockout mice. Triangles represent arterioles, squares represent glomeruli. Closed symbols: *Tie2^fl/fl/Cre-^* mice (n = 8); open symbols: *Tie2^ΔE9^* mice (n = 7).

intra-organ differences. The phenotype of this model provides opportunities, as well as challenges, for future *in vivo* studies on how absence of Tie2 affects endothelial behavior.

Previously, the Dumont lab reported a $Tie2^{+/-}$ heterozygous mouse model based on the deletion of exon 1 [22]. In our quest to take the model one step further and make an inducible, endothelial specific knockout mouse, we started with flanking exon 9 with loxP sites, as deletion of E9 was likely to result in an 'out of frame' mutation and consequent stop codon. At the same time, taking E9 as target for deletion prohibits loxP sites to potentially interfere with the Tie2 promotor region in control conditions, in the absence of Cre-recombinase.

Two recent studies on inducible Tie2 knockout mouse models reported on the location of Tie2 knockout, but focused on non-endothelial Tie2 expression using *Ng2* and *Nfatc1* as Cre promotors specific for pericytes and myocardial cells, respectively [28, 29]. A prominent feature of our endothelial-specific *Tie2^ΔE9^* knockout model is the diverse pattern and extent of Tie2 knockout in the microvasculature of organs, including the absence of knockout in liver. Previously, Göthert et al. showed absence of LacZ staining in liver sinusoids in the *end-SCL-Cre-ER^T^;R26R* reporter mouse, suggesting that Cre-recombinase was either not expressed, or not activated in this vascular compartment [24]. Intriguingly, genomic PCR analysis in our model did reveal deletion of *Tie2* exon 9 in the liver of *Tie2^ΔE9^* knockout mice, indicating effective exposure of Cre-recombinase-expressing endothelial cells to tamoxifen.

An important observation in our *Tie2^ΔE9^* knockout model is that reduction of Tie2 did not exceed 70–85% of control values in whole organs or in specific microvascular compartments. The residual 15–30% Tie2 mRNA and protein is full length Tie2 produced, by cells in which deletion of exon 9 did not take place. This level of knockout is comparable to the levels achieved using the *end-SCL-CRE-ER^T^* mouse for endothelial-specific knockout of other genes. For example, Woo et al. reported 65% *Tie1* knockout in aorta, while Schmidt et al. and Liu et al. reported approximately 60% knockout of *FAK* and *Sox17* in lungs [30–32]. It is highly unlikely that the 15–30% residual Tie2 is derived from Tie2-expressing leukocytes, considering that expression levels of Tie2 in leukocytes are approximately 10-fold lower than in, for example, mouse kidney [23]. Another explanation for the partial reduction of Tie2 levels in our knockout mouse could be the restricted availability of tamoxifen. However, half-life values of tamoxifen and its active metabolite 4OH-tamoxifen in mice are 11.9 hours and 6 hours, respectively [33], suggesting that endothelial cells were extensively exposed to the drug and its metabolite using our administration protocol. Moreover, the reported half-life of Tie2 protein of 9 hours in HUVEC indicates that the experimental time frame chosen in our study was in theory sufficient to obtain a full *Tie2* knockout [34]. A further explanation for the partial knockout of Tie2 could be that the expression of Cre-recombinase differs between endothelial cells or subsets thereof. Furthermore, the accessibility of *Tie2* DNA for Cre-recombinase may vary between endothelial subsets, as recently suggested for immune cell-specific knockout models [35, 36].

Combining *Tie2^floxed/floxed^* transgenic mice with *end-SCL-Cre-ER^T^* transgenic mice resulted in variable, yet highly reproducible Tie2 knockout which was dependent on the location of the endothelial cells in the microvasculature in organs. Future studies will assess whether crossing

of the *Tie2^floxed/floxed* mouse with e.g., *Cdh5Cre^ERT2* or *Flk-1-Cre* mice [37, 38] yields knockout of Tie2 to the same extent and in a similar pattern as we report here. Knocking out another gene in the same Cre-driver mouse may result in different patterns of target gene loss [35, 36, 39]. Important to note is that many studies rely on reporter models to demonstrate where Cre recombinase is active, show target gene knockout in an organ which is not the organ under investigation for functional studies, or only show target gene knockout by *in vitro* exposure to tamoxifen. Information about the extent and location of target gene knockout is the corner-stone of these kinds of studies to safeguard validity of the conclusions drawn, especially since extensive molecular heterogeneity exists between endothelial cells [40, 41] which can have major functional consequences [42].

Zooming in on the microvasculature of the kidney we found a discrepancy between unaltered *Tie2* mRNA expression, and strongly reduced Tie2 protein expression in the arteriolar compartment of knockout mice. At present, this finding cannot be molecularly explained. Possibly, smooth muscle cells that support renal arterioles express *Tie2* mRNA (comparable to Tie2 expression by pericytes in mice [28], and Tie2 expression in smooth muscle cells in rat aorta [43]), yet do not translate this into protein (Fig 3A) due to the presence of microRNAs that can silence Tie2 protein formation [44]. Since laser microdissection enriches samples for endothelial cells but never yields a completely pure endothelial sample, *Tie2* mRNA produced by smooth muscle cells may have contributed to the *Tie2* mRNA signal in these samples. Such an explanation, however, remains hypothetical as Tie2 expression by smooth muscle cell in arterioles in mouse kidney has not been reported. Mechanisms of Tie2 protein expression regulation within specific vascular compartments within an organ require further investigation.

Comparing our results with those of other inducible, endothelial-specific Tie2 knockout mouse models is difficult, since only few such models have been reported so far. In the *Cdh5Cre^ERT2* model, expression of Cre-recombinase was driven by the *VE-cadherin* promoter, and tamoxifen treatment induced deletion of *Tie2* exon 1 [37]. As a result, *Tie2* mRNA in lungs of postnatal d6 mice was reduced by approximately 30%, while in *Tie1/Tie2* double knockout mice approximately 55% less *Tie2* mRNA was reported in lungs at this time point. Information on the exact location of Tie2 knockout in the lung and retinal microvasculature were not provided. An elegant study employing the same mouse model to investigate whether Tie2 exerts a role in tissue repair after photothrombotic injury evoked in the brain of 8 week old mice showed 85% reduction of Tie2 protein expression in the brain [45]. It may well be that also in this *Cdh5Cre^ERT2* mouse model the extent of knockout of Tie2 depends on the organ.

Although not our main focus, we noticed extensive differences between organs in expression of Tie2, its functional facilitator Tie1, and its ligands Ang1 and Ang2. *Tie2* expression in kidney was approximately 4-fold higher than in the other organs. At the same time, the renal microvasculature expressed significantly more *Ang1* and *Ang2* than the microvasculature of other organs, which was accompanied by a *Tie2/Tie1* mRNA ratio of more than 2.5-fold higher than in heart and aorta (S2 and S3 Tables). These noteworthy findings raise questions about the consequences of these differences for (micro)vascular engagement in (patho)physiology. If the model described by Korhonen et al. in trachea and lung of *Tie1* knockout mice were applied to renal vasculature, it would be tempting to speculate that the low *Tie1* relative to *Tie2* levels in kidney would be associated with reduced Tie2 phosphorylation and Akt signaling under homeostatic conditions [7]. In addition, the *Ang1/Ang2* mRNA ratio in the kidney was lower than in heart (>10-fold) and lung (>5-fold) (S2 and S3 Tables), with also yet unknown consequences for Tie2 phosphorylation. The high glomerular expression of members of the Ang/Tie system, including a relatively high expression of *Ang2* ([26] and this paper), and high expression of VEGF and VEGFR2 [18, 26, 46], suggest that in the healthy kidney, glomerular

endothelial cells are set to proliferate. However, in practice, the opposite is true, as exemplified by the lack of glomerular endothelial cell proliferation in healthy mouse kidney, and the poor *in vitro* proliferation of isolated mouse glomerular endothelial cells [47]. Similarly enigmatic is the observation that in glomeruli of transgenic mice, the balance in Tie2 /Tie1 expression, such as is present under control conditions, is altered. No such alterations were present in arterioles. Methods that enable the localization and quantitation of the four proteins of the Ang/Tie family studied here, and that can be combined with quantitation of Tie2 phosphorylation in discrete microvascular segments, will be crucial for elucidating how these molecular systems contribute to heterogeneous microvascular endothelial behavior.

In summary, we generated an endothelial-specific, inducible *Tie2* knockout mouse model with highest knockout in lung, aorta, and heart. Upon zooming in on microvascular compartments, Tie2 knockout was found to be extensive in arterioles of kidney, lung, and heart, and in lung capillaries. While this heterogeneous knockout pattern can be seen as a drawback of the model, it is a common phenomenon in inducible (endothelial) knockout models [39]. This variation in knockout extent requires a meticulous analysis of the knockout status of the gene of interest prior to studying its role in pathological conditions. Our study of the Tie2 molecule illustrates why it would be insufficient and even counterproductive to restrict confirmation of knockout to the vasculature of a single organ or tissue, when the aim is to investigate its role in another organ or tissue. Conversely, the extensive Tie2 knockout extent in lung, aorta, and heart, and in the arterioles of all organs, provides unique opportunities to investigate the role of the Tie2 system in these mature microvascular beds in health and disease. Combining our new *Tie2$^{floxed/floxed}$;end-SCL-Cre-ER$^{T}$* Tie2 knockout model with (patho)physiological models for studies into the biological and functional relevance of low/absent Tie2 expression in specific blood vessels, is within reach, yet requires attention to be paid to the extent and location of knock out in the natural organ context.

## Supporting information

**S1 Fig. Confirmation of *Tie2* exon 9 deletion in organs of *Tie2* knockout mice.** After tamoxifen-induced activation of Cre-recombinase, the presence (a 940 base pair (bp) PCR product) or absence (a 309 bp PCR product) of *Tie2* exon 9 was confirmed by genomic PCR in kidney, lung, heart, aorta, and liver tissue of *Tie2$^{ΔE9}$* knockout mice (n = 7) and *Tie2$^{fl/fl/Cre-}$* control mice (n = 8). Lanes show genomic PCR products for individual mice. Kidneys and lungs were genotyped in two separate groups of mice, hence PCR products are shown on separate gels. (TIF)

**S2 Fig. Body weights of control and *Tie2* knockout mice during and after tamoxifen treatment.** *Tie2$^{fl/fl/Cre-}$* control mice and *Tie2$^{ΔE9}$* knockout mice were intraperitoneally injected with tamoxifen (4mg/injection), three times a week for a period of three weeks. Body weights were measured at 2-3-day intervals and normalized to the weights at the start of the experiment. Symbols represent mean values ± SD (error bars). Closed circles with closed line: *Tie2$^{fl/fl/Cre-}$* mice (n = 8); open circles with dashed line: *Tie2$^{ΔE9}$* mice (n = 7). (TIF)

**S3 Fig. Expression of *VE-cadherin* and *CD31* in control and *Tie2* knockout mice.** mRNA expression of the pan-endothelial genes *VE-cadherin* and *CD31* was quantified in organs by RT-qPCR. (**A**) Gene expression of *VE-cadherin* was normalized to the housekeeping gene *Gapdh*. (**B**) Gene expression of *CD31* was normalized to *VE-cadherin*. Graphs show individual values and means (black lines) ± SD (error bars). Closed circles: *Tie2$^{fl/fl/Cre-}$* control mice

(n = 8); open circles: $Tie2^{\Delta E9}$ knockout mice (n = 7).
(TIF)

**S4 Fig. *Tie2* mRNA expression in lungs of control and *Tie2* knockout mice determined using 3 different primer/probe sets.** *Tie2* mRNA expression levels were determined in lungs of $Tie2^{fl/fl/Cre-}$ control mice and $Tie2^{\Delta E9}$ knockout mice after tamoxifen treatment by RT-qPCR using 3 different primer/probe sets that bind at the boundary of exon 7/8, exon 8/9, or exon 15/16 of *Tie2*. (**A**) *Tie2* expression in $Tie2^{fl/fl/Cre-}$ control, and $Tie2^{\Delta E9}$ knockout samples. Exon binding sites of the primer/probe sets are indicated under the X-axis. Graph shows individual values and means (black lines) ± SD (error bars). Closed circles: $Tie2^{fl/fl/Cre-}$ mice (n = 8); open circles: $Tie2^{\Delta E9}$ mice (n = 7) (**B**) Percent knockout of *Tie2* mRNA in $Tie2^{\Delta E9}$ knockout samples relative to control mice. Bars shows means ± SD (error bars).
(TIF)

**S1 Table. Comparison of *Tie1*, *Tie2*, *Ang1*, and *Ang2* mRNA expression levels in organs of control mice.** Differences in mRNA levels of *Tie1*, *Tie2*, *Ang1*, and *Ang2* (Fig 4) between organs of $Tie2^{fl/fl/Cre-}$ control mice were determined by Sidak's multiple comparisons test. Adjusted P values are reported.
(DOCX)

**S2 Table. *Tie2/Tie1* and *Ang1/Ang2* mRNA expression ratios in organs of control mice.** Calculated ratios of mRNA expression levels in organs of $Tie2^{fl/fl/Cre-}$ control mice as presented in Fig 4.
(DOCX)

**S3 Table. Comparison of *Tie2/Tie1* and *Ang1/Ang2* mRNA expression level ratios in organs of control mice.** Differences in mRNA level ratios of *Tie2/Tie1* and *Ang1/Ang2* of $Tie2^{fl/fl/Cre-}$ control mice (S2 Table) were determined by Sidak's multiple comparisons test. Adjusted P values are reported.
(DOCX)

**S1 Raw images. Pdf file containing original TIFF images of the electrophoreses gels used for Figs 1, 5 and S1 Fig.** Each raw image is labeled and annotated to identify corresponding bands used in the manuscript figures by use of Adobe Photoshop v22.1.1.
(PDF)

**S1 Dataset. Excel data file that represents all raw data points of the Tie2 ELISA used for Fig 2B.** Standard curve is shown for samples measured in one run.
(XLSX)

**S2 Dataset. GraphPad data file used for the statistical analysis of mRNA expression levels as presented in Figs 2A, 4A–4C, and S1 Table.**
(PZFX)

**S3 Dataset. GraphPad data file used for the statistical analysis of Tie2 protein levels (ELISA) as presented in Fig 2B.**
(PZFX)

**S4 Dataset. GraphPad data file used for the statistical analysis of Tie2 protein localization in different organs (IHC) by morphometrics as presented in Fig 3B.**
(PZFX)

**S5 Dataset. GraphPad data file used for the statistical analysis of mRNA expression levels in laser microdissected compartments as presented in Fig 5B–5E.**
(PZFX)

**S6 Dataset. GraphPad data file used for the statistical analysis of Tie1-Tie2 correlations in laser microdissected compartments as presented in Fig 5F, 5G.**
(PZFX)

**S7 Dataset. GraphPad data file used for the statistical analysis of VE-cadherin and CD31 mRNA expression levels as presented in S3 Fig.**
(PZFX)

**S8 Dataset. GraphPad data file used for the statistical analysis of *Tie2*/*Tie1* and *Ang1*/*Ang2* mRNA level ratios as presented in S2 and S3 Tables.**
(PZFX)

**S9 Dataset. GraphPad data file used for the statistical analysis of *Tie2* mRNA expression levels using 3 different primer/probe sets as presented in S4 Fig.**
(PZFX)

## Acknowledgments

The authors would like to thank Bart van der Sluis for technical advice and Henk Moorlag and Arjen Petersen for technical assistance. Furthermore, we would like to thank the staff of the UMCG animal facility for taking care of the mice and technical assistance during the experimental part of this study. Lastly, we thank the Telethon KIDS Institute, Western Australia, for kindly providing the *end-SCL-Cre-ER^T* transgenic mice.

## Author Contributions

**Conceptualization:** Peter J. Zwiers, Rianne M. Jongman, Radu V. Stan, Joachim R. Göthert, Eliane R. Popa, Grietje Molema.

**Data curation:** Peter J. Zwiers, Rianne M. Jongman, Timara Kuiper, Jill Moser, Matijs van Meurs, Eliane R. Popa, Grietje Molema.

**Formal analysis:** Peter J. Zwiers, Rianne M. Jongman, Timara Kuiper, Eliane R. Popa, Grietje Molema.

**Funding acquisition:** Grietje Molema.

**Investigation:** Peter J. Zwiers, Rianne M. Jongman, Timara Kuiper, Jill Moser, Matijs van Meurs, Eliane R. Popa, Grietje Molema.

**Methodology:** Peter J. Zwiers, Rianne M. Jongman, Jill Moser, Radu V. Stan, Joachim R. Göthert, Matijs van Meurs, Eliane R. Popa, Grietje Molema.

**Resources:** Radu V. Stan, Joachim R. Göthert.

**Software:** Peter J. Zwiers, Eliane R. Popa.

**Supervision:** Eliane R. Popa, Grietje Molema.

**Visualization:** Peter J. Zwiers, Rianne M. Jongman, Eliane R. Popa, Grietje Molema.

**Writing – original draft:** Peter J. Zwiers, Eliane R. Popa, Grietje Molema.

**Writing – review & editing:** Peter J. Zwiers, Rianne M. Jongman, Timara Kuiper, Jill Moser, Radu V. Stan, Joachim R. Göthert, Matijs van Meurs, Eliane R. Popa, Grietje Molema.

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
