## [Decision Letter · Decision Letter 0]

8 Mar 2021

PONE-D-21-04716

Pattern of inducible endothelial Tie2 knockout in mature mouse microvasculature is organ- and vascular compartment-dependent

PLOS ONE

Dear Dr. Zwiers,

Thank you for submitting your manuscript to PLOS ONE. After careful consideration, we feel that it has merit but does not fully meet PLOS ONE’s publication criteria as it currently stands. Therefore, we invite you to submit a revised version of the manuscript that addresses the points raised during the review process.

We look forward to receiving your revised manuscript.

Kind regards,

Slava Rom, Ph.D.

Academic Editor

PLOS ONE

Journal Requirements:

2.PLOS ONE now requires that authors provide the original uncropped and unadjusted images underlying all blot or gel results reported in a submission’s figures or Supporting Information files. This policy and the journal’s other requirements for blot/gel reporting and figure preparation are described in detail at https://journals.plos.org/plosone/s/figures#loc-blot-and-gel-reporting-requirements and https://journals.plos.org/plosone/s/figures#loc-preparing-figures-from-image-files. When you submit your revised manuscript, please ensure that your figures adhere fully to these guidelines and provide the original underlying images for all blot or gel data reported in your submission. See the following link for instructions on providing the original image data: https://journals.plos.org/plosone/s/figures#loc-original-images-for-blots-and-gels.

Reviewers' comments:

Reviewer's Responses to Questions

**Comments to the Author**

1. Is the manuscript technically sound, and do the data support the conclusions?

Reviewer #1: Partly

Reviewer #2: No

Reviewer #3: Yes

2. Has the statistical analysis been performed appropriately and rigorously? 

Reviewer #1: No

Reviewer #2: I Don't Know

Reviewer #3: Yes

3. Have the authors made all data underlying the findings in their manuscript fully available?

Reviewer #1: Yes

Reviewer #2: Yes

Reviewer #3: Yes

4. Is the manuscript presented in an intelligible fashion and written in standard English?

Reviewer #1: Yes

Reviewer #2: Yes

Reviewer #3: Yes

5. Review Comments to the Author

Reviewer #1: PONE-D-21-04716

The manuscript “Pattern of inducible endothelial Tie2 knockout in mature mouse microvasculature is organ- and vascular compartment-dependent” aimed at the description of novel inducible Tie2 KO mouse model. Authors reported successful deletion of Tie2 in various organ and vascular compartments. Despite the fact that Tie2 KO models bear potential interest and are important for the field of endothelial pathobiology, the presented manuscript has significant weaknesses such as incomplete description of results and lack of proper discussion of the reported findings. Therefore, I cannot recommend it for publication in its current state.

Major

Why was brain Tie2 expression upon knockout not examined?

Protein levels of Tie1 have to be determined as well. Especially considering quite small degree of mRNA reduction. What is justification behind of selective description of kidney Tie1 mRNA compartmentalization (Figure 5)? Why are other organs not analyzed?

Discussion section does not provide proper analysis of the reported findings.

Statistics and Results:

Given the small sample size, were the samples normally distributed? What normality test was used?

Result section is incomplete (no information about “means ± SEM or SD” whatsoever).

Figures (plots) do not have any statistical info embedded (no SEM or SD bars)

Minor

Introduction:

First paragraph of introduction (lines 42 – 56) would benefit from editing to correct logical and grammar flaws.

Results/Figures:

Panels in Figure 3 need to have scale bars and magnification indicators.

Reviewer #2: In this report, the authors have used their recently developed Tie2 exon 9-floxed mouse line to generate a conditional, endothelial cell (EC)-specific Tie2 knockout and to examine effects on expression of Tie2 and the angiopoietins in various tissues. The aim of the study, as stated in line 62, was to generate and characterize this model to allow future studies of Tie2. The results suggest that there is heterogeneous knockout of Tie2 in various tissues in this model. Based on these results the authors suggest that a thorough analysis of the knockout patterns of specific genes be investigated before investigating their role in pathophysiological conditions. This is a very reasonable statement, and this study does that to a certain extent. However, there is an important concern about the line itself, most notably that the deletion of exon 9 alone might lead to residual mRNA and/or protein expression, which may confound the authors’ observations. Furthermore, the Cre driver used here (SCL) has been shown to be relatively inefficient, as acknowledged by the authors, so it is not entirely surprising that Tie2 deletion was heterogeneous. However, this raises some concern about using this model for pathophysiological studies, which is what this characterization purportedly sets the stage for. Specific comments are listed below.

Major comments:

1. Validity of the model for Tie2 deletion – The apparent lack of complete Tie2 deletion of in ECs raises doubt about the mouse model. The deletion of exon 9 potentially averts the possibility of interfering with promoter/enhancer elements near the 5’ end of the gene. However, it is possible that this targeting strategy does not completely delete Tie2. If not, it may be detected by the Taqman PCR primer pair used to analyze Tie2 expression, which targets the exon 7-8 boundary. Furthermore, there may be residual protein expression, which may be detected by an antibody directed against the extracellular domain of Tie2, which TEK4 and the R&D ELISA appear to do. Without addressing this issue, it is difficult to draw any conclusions about the authors observations regarding effects on expression of Tie2 and the other Tie family genes/proteins.

The discrepant finding that Tie2 mRNA is persistently expressed in arterioles (Fig. 5B) whereas protein expression is not (Fig. 3A) leads the authors to invoke miRNAs, when in fact the explanation may be much simpler, as noted above. Moreover, it is not clear why miRNAs would be upregulated preferentially in the knockout vs. the wild-type mouse.

To address these issues, it will be important to test another primer pair downstream of the deletion, and it would also be important to demonstrate absence of protein expression by western blotting of tissues (or isolated cells) to determine whether a truncated protein product is expressed as a result of the deletion.

2. Relative inefficiency of the SCL-Cre-ERT driver – The apparent “heterogeneity” of Tie2 deletion may just be due to poor excision from this Cre driver, which has been shown to be as low as 60% efficient. The fact that Hprt-Cre-mediated deletion of Tie2 in the authors’ earlier paper resulted in embryonic lethality suggests that the targeting approach works. Thus, poor deletion with this Cre driver might result in a poor model for any pathophysiological studies. Although such pathological models may be beyond the scope of this paper, it is easy to conclude that this model might have limited efficacy without demonstration otherwise. The use of a reporter line, such as an SCL-Cre-ERT;ROSA-TdTomato line, to demonstrate simultaneously where the Cre is being expressed along with residual Tie2 co-expression would be helpful. This is particularly relevant in the kidney, where there is apparently such high residual Tie2 expression.

3. Effects of Tie2 deletion on expression of other genes – The data suggest that expression of Tie1, Ang1, and Ang2 is variably altered by Tie2 deletion. However, the veracity of these data is not clear given the questions about residual Tie2 expression, and this impacts the correlations in Fig 5C. Moreover, many of the statistically significant changes in gene expression appear trivial, such as the 12% reduction in Tie1 mRNA and the 17% increase in Ang2 expression in Tie2-deficient hearts (Fig 4A,C), thus it is not clear whether these changes have any biological significance. Without testing this in some physiological or pathophysiological model, as suggested in point #2, the relevance of these findings remains unclear.

Minor comments:

1. Tie2 nomenclature (line 42) – The original name for Tie2 was Tunica interna Endothelial cell Kinase, or TEK (still the primary gene name) (Dumont, Oncogene, 1992). If the authors wish to acknowledge any name besides TIE2, it is suggested that this names be used, as “Tyrosine kinase receptor 2” is generic and not an accepted alias, although the related “Tyrosine-Protein Kinase Receptor TIE-2” is listed in GeneCards.

Reviewer #3: This paper reports the pattern of inducible KO of tie2 using a particular floxed allele of tie2 (exon 9 deletion) and a particular tamoxifen inducible- Cre driver (end-SCL-Cre-ERT). I state this up front since any conclusions drawn from this study are primarily dependent on the choice of the Cre driver and to a lesser extent the choice of the deleted exon in the tie2 gene (that is how easily the Cre can perform the KO given the genomic sequence context and distance between the flox sites).

But with that caveat the work is very meticulous and robust. The main conclusions of heterogenous organ and tissue specific KO are valuable, and also, and possibly more importantly, as a reference and a warning for really any inducible KO system for any gene. I fully trust their data.

Suggestion: From what I gather from the Methods, the animals were injected with TMX 3 times, once each in 3 successive weeks, at the dose listed, but at what age did the injections start? Whether this is an adult-age deletion or early-postnatal is critical to all of the results, but this information was not given. But this information has to be clearly stated in Methods. And equally importantly, since, the conclusions drawn depend on the age of deletion, the Discussion include comments on what this means in light of whether this was adult or early postnatal deletion (or other).

6. PLOS authors have the option to publish the peer review history of their article (what does this mean?). If published, this will include your full peer review and any attached files.

Reviewer #1: No

Reviewer #2: No

Reviewer #3: No

---

## [Author Response · Author response to Decision Letter 0]

2 Sep 2021

Rebuttal for PONE-D-21-04716: “Pattern of inducible endothelial Tie2 knockout in mature mouse microvasculature is organ- and vascular compartment-dependent” by Zwiers et al.

We would like to thank the reviewers for their detailed and insightful comments, which we will address below in a point-to-point manner. We also would like to thank the editor for giving us the opportunity to respond to these comments. Due to ongoing COVID-19-related lab restrictions, temporary lack of plastics, and delays in delivery of materials due to Brexit, we needed extended time for the execution of some of the new analyses, the subsequent writing of the rebuttal, and revising the manuscript. We sincerely apologize for this delay.

Tables and Figures included in this rebuttal are marked R and are for rebuttal purposes only, unless otherwise stated. Lines to which we refer can be found in the cleaned-up version of the revised manuscript.

Reviewer #1.

Comment Rev1-1: Why was brain Tie2 expression upon knockout not examined?

Response to Rev1-1: The larger interest of our research revolves around endothelial heterogeneity in organs that are affected by systemic inflammatory processes as prevail in sepsis, endotoxemia, and shock induced by hemorrhage. In previous research, we and other showed that in animal models of these conditions, organ dysfunction and Tie2 loss occurs mainly in lung, kidney, and liver, while the heart and aorta are considered resistant to inflammation and vascular permeability. Indeed, brain can also be considered an organ of interest, yet the vessels in the brain express lower levels of Tie2 compared to the organs reported here. We furthermore lack expertise to analyze the individual components of this large and anatomically complex organ. This led us to exclude this organ from the current study.

Comment Rev1-2: Protein levels of Tie1 have to be determined as well. Especially considering quite small degree of mRNA reduction. What is justification behind of selective description of kidney Tie1 mRNA compartmentalization (Figure 5)? Why are other organs not analyzed?

Response to Rev1-2: In our hands, small changes in mRNA levels of genes of interest was never associated with quantitatively picking up changes in protein levels when using Western Blot. In addition, we were not able to find commercial ELISA kits for Tie1 quantitation that were validated for analysis of tissue homogenates. Similarly, while considering IHC/IF worthwhile pursuing, as these techniques may also enable identification of possible heterogeneous expression loss of Tie1, we have not been able to find suitable antibodies for this purpose. 

As we agree with the reviewer that quantitative localization of Tie1 and the other members of the Ang/Tie system will improve our understanding of local changes in expression in response to (partial) loss of Tie2, we have adapted the Discussion section as follows (Line 404):

Methods that enable the localization and quantitation of the four proteins of the Ang/Tie family studied here, and that can be combined with quantitation of Tie2 phosphorylation in discrete microvascular segments, will be crucial for elucidating how these molecular systems contribute to heterogeneous microvascular endothelial behavior.

Concerning the “selective description of kidney Tie1 mRNA compartmentalization (Figure 5)”: We assume that the reviewer is asking why only arterioles and glomeruli were analyzed. Owing to the physical limitations posed by the size of the laser beam used for Laser Microdissection (LMD), we are able to dissect sufficient numbers of arterioles and glomeruli for subsequent RT-qPCR analyses. However, LMD of microvascular beds from heart and liver sections has so far not provided sufficient material for further analysis. Finally, LMD of microvascular beds from lung sections is virtually impossible, since lung tissue harvested in this study collapsed during excision and snap-freezing.

From our perspective, introducing the (limited) information on two kidney microvascular segments is still valuable for readers, as Tie2 is heavily investigated as target for therapeutic intervention in acute kidney injury (AKI).

Question Rev1-3: Discussion section does not provide proper analysis of the reported findings.

Response to Rev1-3: We apologize for the fact that we do not fully understand the core criticism of this comment. If the reviewer refers to the fact that we did not discuss the data per figure: our approach is, and has been over the years, to highlight the most important findings in relation to findings published by others, and to discuss methodological issues and cell biological observations that we considered to be of interest for the readers.

In this rebuttal we address some issues that have now been incorporated in the Discussion section of our revised manuscript. Possibly, some of these represent what reviewer 1 identified as lacking.

Question Rev1-4: Given the small sample size, were the samples normally distributed? What normality test was used? Result section is incomplete (no information about “means ± SEM or SD” whatsoever).

Figures (plots) do not have any statistical info embedded (no SEM or SD bars). 

Response to Rev1-4:

We thank the reviewer for this remark, as indeed, the normality test was not mentioned in the original manuscript. We performed a Shapiro-Wilk normality test on mRNA expression data in whole organ extracts and in LMD isolates from the kidney (see Table R1). We inserted a clarifying sentence at line 148-150 in the original manuscript: 

Normality of distribution was tested using the Shapiro-Wilk normality test on Tie2 mRNA data obtained from whole organ extracts and LMD isolates from the kidney. The results showed that values are normally distributed.

Table R1: Shapiro-Wilk normality test for Tie2 mRNA level sample distribution in whole organ isolates and glomeruli and arterioles laser microdissected from kidney.

As suggested by the reviewer, we added SD bars and corresponding information to figures 2, 3B, 4, 5, S2, S3, and S5. As we provide information on means and p-values in the legends, and now also SD values in the figures, we consider mentioning this information in the Results section as redundant. 

Minor questions Reviewer 1: 

Rev1-5: Introduction: First paragraph of introduction (lines 42 – 56) would benefit from editing to correct logical and grammar flaws. 

Response to Rev1-5: Based on this comment we changed the first paragraph as follows (line 43-54): 

Tyrosine-protein kinase receptor Tie2, also known as Tunica interna Endothelial cell Kinase, or TEK, is expressed by vascular endothelial cells, as well as other cell subsets of the hematopoietic lineage [1,2]. Endothelial Tie2 plays a prominent role in blood vessel development, vascular integrity, and endothelial responses to inflammation [3–6]. The current model of endothelial Tie2 engagement in signal transduction describes binding of non-endothelially produced agonist angiopoietin (Ang)-1 to Tie2 under quiescent conditions(7, 8). Binding of Ang-1 induces autophosphorylation of Tie2, resulting in PI3-Akt signal transduction and blood vessel stabilization, ensuring vascular integrity [9-–13]. In acute inflammation, such as prevails in, for example, sepsis, endothelially stored Ang-2 is rapidly released from Weibel-Palade bodies and competes with Ang-1 for binding to Tie2 [14–17]. As a result, Tie2 activation is aborted, and endothelial barrier function and vascular integrity are lost. Tie2 signaling is not only controlled by a balance between levels of Ang-1 and Ang-2, but also by heterodimerization of Tie2 with Tyrosine-protein kinase receptor Tie1, which stabilizes Tie2 auto-phosphorylation and downstream signaling [7]. 

Rev1-6: Results/Figures: Panels in Figure 3 need to have scale bars and magnification indicators.

Response to Rev1-6: Due to the large number and small size of the photomicrographs in Figure 3A, we were not able to add scale bars to these figures in a legible manner. Instead, we added information about the magnifications used to make the photomicrographs in the legend of Figure 3A (line 246).

Reviewer #2

For clarity, we divided the major criticism into two separate parts. The first part deals with the issue whether truncated Tie2 mRNA is produced when exon 9 is deleted (creating the ΔE9 transgenic). The second part addresses the status of Tie2 protein after exon 9 deletion.

Question Rev2-1: Validity of the model for Tie2 deletion – The apparent lack of complete Tie2 deletion in ECs raises doubt about the mouse model. The deletion of exon 9 potentially averts the possibility of interfering with promoter/enhancer elements near the 5’ end of the gene. However, it is possible that this targeting strategy does not completely delete Tie2. If not, it may be detected by the Taqman PCR primer pair used to analyze Tie2 expression, which targets the exon 7-8 boundary. […] To address these issues, it will be important to test another primer pair downstream of the deletion.

Response to Rev2-1: We thank the reviewer for this comment. In all honesty, we never considered that a truncated Tie2 product could be formed upon excision of exon 9, as in silico assessment of the consequences of deletion of exon 9 of Tie2 showed a high probability of abrogation of Tie2 mRNA synthesis and abrogation of protein formation. Moreover, in our previous publication (Jongman et al, Shock (51); 6; p757-769) we reported the construction of Tie2+/- heterozygous mice, for which we crossed the same exon 9 Tie2floxed/floxed mice with Hprt-Cre mice. This transgenic model was characterized by a 50% reduction in Tie2 mRNA and protein in various organs. Furthermore, F1xF1 litter only consisted of wildtype and Tie2+/- mice, while homozygous Tie2-/- mice were not viable. These findings indicate loss of Tie2 upon exon 9 deletion. 

Considering the importance of this reviewer’s remarks, we pursued the advice and performed RT-qPCR analyses with two additional Taqman primer/probe sets recognizing Tie2 mRNA at exon boundary 8/9, respectively 15/16. If upon exon 9 deletion truncated Tie2 mRNA were formed, one would expect similar levels of mRNA in Tie2fl/fl/Cre- control and Tie2∆E9 samples using primer/probe set 7/8. Primer/probe sets 8/9 and 15/16 would only yield product in Tie2fl/fl/Cre- control samples, in which full length Tie2 mRNA is present. 

Figure R1-A shows that the extent of Tie2 mRNA product formation by the qPCR reaction differed between the 3 primer/probe sets. When applying them to Tie2∆E9 transgenic mouse lungs, all primer/probe sets showed approximately 50% reduction in Tie2 mRNA compared to control mice (figure R1-B). From these data, which we have now introduced in the revised version of the manuscript as S5 Fig, we conclude that exon 9 deletion does not yield truncated Tie2 mRNA. Tie2 mRNA detected in Tie2∆E9 mice with the exon 7/8 primer/probe set used throughout this study, therefore only represents full length Tie2 mRNA. This can be considered residual and points to heterogeneous knock out of Tie2 in different locations of the microvasculature.

Figure R1. Tie2 mRNA expression in lungs of control and Tie2 knockout mice determined by use of 3 different primer/probe sets. Tie2 mRNA expression levels were determined in lungs of Tie2fl/fl/Cre- control mice and Tie2�E9 knockout mice after tamoxifen treatment by RT-qPCR and use of 3 different primer/probe sets that bind at the boundary of Tie2 exon 7/8, exon 8/9, or exon 15/16. (A) Tie2 expression in Tie2fl/fl/Cre- control, and Tie2�E9 knockout samples. Exon binding sites of the primer/probe sets are indicated under the X-axis. The graph shows individual values and means ± SD. Closed circles: Tie2fl/fl/Cre- mice (n=8); open circles: Tie2�E9 mice(n=7) (B) Percent knockout of Tie2 mRNA in Tie2�E9 knockout samples relative to control mice. Bars shows means ± SD. 

The following revisions have been made in the manuscript:

In the Materials and Methods section we inserted relevant exon 8/9 and exon 15/16 primer/probe set information in Table 1 (RT-qPCR primers), line 113:

Tek Mm00443243_m1 Tyrosine-protein kinase receptor (Tie2), CD202 (exon 8/9)

Tek Mm01256894_m1 Tyrosine-protein kinase receptor (Tie2), CD202 (exon 15/16)

We also added the following sentence (line 103):

To determine the possible formation of truncated Tie2 mRNA upon exon 9 deletion, we used 3 different primer/probe sets that hybridize at cDNA sites before (at the boundary between exon 7 and exon 8, hereafter referred to as ‘exon 7/8’), at (exon 8/9), or after exon 9 (exon 15/16) (see Table 1).

Question Rev2-2: Furthermore, there may be residual protein expression, which may be detected by an antibody directed against the extracellular domain of Tie2, which TEK4 and the R&D ELISA appear to do. Without addressing this issue, it is difficult to draw any conclusions about the authors observations regarding effects on expression of Tie2 and the other Tie family genes/proteins.

Response to Rev2-2: The Tie2 antibody used for IHC staining indeed detects the extracellular domain. Although R&D systems does not reveal the exact identity of the capture antibody in its ELISA kit (Mouse Tie-2 Immunoassay, MTE200 of R&D Systems), the fact that the ELISA is widely used to quantitate soluble Tie2, makes it safe to conclude that it also detects the extracellular domain of Tie2. 

Although highly unlikely, as discussed in Response to Rev2-1, if exon 9 deletion resulted in the formation of a truncated Tie2 protein, this protein would only comprise the extracellular domain, and thus would be much smaller than the full length protein. To investigate whether such smaller Tie2 proteins had formed in the transgenic mice, we subjected lung homogenates of Tie2fl/fl/Cre and Tie2∆E9 mice to Western Blot. Proteins were detected using two antibodies: Ab33 from Upstate detects the extracellular domain of Tie2, and an Ab from Cohesion Bioscience recognizes Tie2 p-tyrosine at position 1108, i.e., the intracellular domain. Although smaller bands are visible at ~100kD, 45kD, and 20kD, these bands are not restricted to the samples of the Tie2∆E9 mice (Figure R2), indicating that they do not represent truncated Tie2 protein formed as a result of deletion of exon 9. 

Figure R2: Determination of the molecular weight of Tie2 protein in mouse lungs by Western Blot. Briefly, lung protein isolates of Tie2fl/fl/Cre-control and Tie2∆9 knockout mice (n=3 per group, 25ug per sample), were run on a 10% SDS gel, proteins were transferred to a nitrocellulose membrane, and detected with Tie2 Ab33 (cat no 05-584, Upstate®, Merck (Sigma Aldrich), Amsterdam, The Netherlands) and Tek C-Term, pY1108 (cat no. ABIN2705294, Cohesion Biosciences, London, UK). 

One could argue that truncated Tie2 protein, since it has lost its transmembrane domain, would not be detectable in organs but rather in a soluble form in plasma. To assess whether this is the case upon exon 9 deletion, we analyzed plasma from Tie2+/- heterozygous mice (referred to above) that were bred by crossing Tie2floxed/floxed exon 9 mice with Hprt-Cre mice. In these Tie2+/- mice, sTie2 levels in plasma did not increase, but instead were 50% lower than those in wildtype littermates (Figure R3), supporting the conclusion that excision of exon 9 of Tie2 does not result in truncated Tie2 protein.

Based on the above, we conclude that Tie2 mRNA and protein detected in the Tie2∆E9 mice reported in our manuscript, represent full length, residual Tie2, rather than truncated mRNA/protein. We have adapted the revised manuscript as follows:

In the Results section (line 204-213):

To confirm that reduced levels of Tie2 originate from residual of full length Tie2 and is not a result of the formation of truncated Tie2, we performed RT-qPCR analysis on lung RNA isolates from Tie2fl/fl/Cre- control mice and Tie2∆E9 knockout mice. For this analysis we used three primer/probe sets that bind to different intron-overspanning regions within the Tie2 cDNA. While they hybridized to the Tie2 cDNA to different extents, the percentage of Tie2 downregulation in the Tie2∆E9 knockout mice versus controls measured by each of these primer/probe sets was approximately 50% and comparable to one another (S5 Fig). We also performed Western Blot analysis on protein isolates from the lung, and did not detect small size, truncated Tie2 protein in Tie2∆E9 knockout mice compared to controls (data not shown). Based on these results we concluded that the decrease in Tie2 mRNA and protein levels in our mouse model represents residual full length Tie2.

In the Discussion, we changed and added the following (line 348): 

An important observation in our Tie2�E9 knockout model is that reduction of Tie2 did not exceed 70-85% of control values in whole organs or in specific microvascular compartments. The residual 15-30% Tie2 mRNA and protein is full length Tie2, produced by cells in which deletion of exon 9 did not take place. 

Question Rev2-3: The discrepant finding that Tie2 mRNA is persistently expressed in arterioles (Fig. 5B) whereas protein expression is not (Fig. 3A) leads the authors to invoke miRNAs, when in fact the explanation may be much simpler, as noted above. Moreover, it is not clear why miRNAs would be upregulated preferentially in the knockout vs. the wild-type mouse…[….]….it would also be important to demonstrate absence of protein expression by western blotting of tissues (or isolated cells) to determine whether a truncated protein product is expressed as a result of the deletion.

Response to Rev2-3: At present, it is technically impossible to quantitate Tie2 protein by Western Blot or ELISA in cells isolated from specific microvascular segments by enzymatic digestion or laser microdissection, due to lack of markers to identify these subsets, respectively limited yield of material in the LMD procedure. The additional data generated and added to the revised manuscript showing that Tie2 mRNA and protein in the Tie2∆E9 knockout mice is full length residual Tie2, made us conclude that right now the main outcome of our study firmly stands, namely that in our new model of TMX-induced endothelial knockout of Tie2, some vessels show a high extent of protein loss, while others are hardly affected - whatever the molecular reason is. 

We agree with the reviewer that our explanation of why in some cases protein levels of Tie2 in dedicated microvascular segments are reduced, while mRNA levels are unchanged, is speculative. We have revised this paragraph as follows, to make this clear (line 368-378):

Zooming in on the microvasculature of the kidney we found a discrepancy between unaltered Tie2 mRNA expression, and strongly reduced Tie2 protein expression in the arteriolar compartment of knockout mice. At present, this finding cannot be molecularly explained. Possibly, smooth muscle cells that support renal arterioles express Tie2 mRNA (comparable to Tie2 expression by pericytes in mice) [Teichert et al, Nat. Commun. 2017;8: 16106], and Tie2 expression in smooth muscle cells in rat aorta [Iurlaro et al, J. of Cell Science 2003;116: 3635-3543], yet do not translate this into protein (Fig 3A) due to the presence of microRNAs that can silence Tie2 protein formation [Besnier et al, Mol. Ther. - Nucleic Acids, 2019;17: 49–62]. Since laser microdissection enriches samples for endothelial cells but never yields a completely pure endothelial sample, Tie2 mRNA produced by smooth muscle cells may have contributed to the Tie2 mRNA signal in these samples. Such an explanation, however, remains hypothetical as Tie2 expression by smooth muscle cell in arterioles in mouse kidney has not been reported. Mechanisms of Tie2 protein expression regulation within specific vascular compartments within an organ require further investigation.

Question Rev2-4: Relative inefficiency of the SCL-Cre-ERT driver – The apparent “heterogeneity” of Tie2 deletion may just be due to poor excision from this Cre driver, which has been shown to be as low as 60% efficient. The fact that Hprt-Cre-mediated deletion of Tie2 in the authors’ earlier paper resulted in embryonic lethality suggests that the targeting approach works. Thus, poor deletion with this Cre driver might result in a poor model for any pathophysiological studies. Although such pathological models may be beyond the scope of this paper, it is easy to conclude that this model might have limited efficacy without demonstration otherwise. The use of a reporter line, such as an SCL-Cre-ERT;ROSA-TdTomato line, to demonstrate simultaneously where the Cre is being expressed along with residual Tie2 co-expression would be helpful. This is particularly relevant in the kidney, where there is apparently such high residual Tie2 expression.

Response to Rev2-4: The choice of the SCL-Cre-ERT driver was based on the publication by Göthert et al. (Blood 2004;104; p1768-1777), who showed high penetrance of Cre expression in the microvasculature of several organs using an SCL-Cre-ERT LacZ reporter mouse. Our investigation aimed to apply this technology to knock out endogenous Tie2. The results presented in this paper lead us to conclude that a direct relation between Cre-based LacZ expression and Cre-based Tie2 knockout cannot be established. Using an additional reporter line, as proposed by the reviewer, is interesting from the point of view that it would provide experimental data explaining the heterogeneous loss of Tie2, yet will not change the outcome of the model. We would like to stress the fact that by now it is broadly accepted that none of the endothelial-specific driver models accomplish 100% knockout in all blood vessels in all organs, as summarized by Payne et al (ATVB 2018; 38; p2550-2561). 

We do not agree with this reviewer’s conclusion that this model might have limited efficacy, since critical analyses of the (micro)vascular beds within the organs studied in this manuscript revealed an 80% reduction in Tie2 expression in several of these. Still, each organ and microvascular bed should be investigated for extent of knockout before any (mechanistic and/or pathophysiology related) conclusions can be drawn. This recommendation not only holds true for SCL-Cre-ERT driver models, but for any model, as we discuss in line 412-416 of the revised manuscript. 

Question Rev2-5: Effects of Tie2 deletion on expression of other genes – The data suggest that expression of Tie1, Ang1, and Ang2 is variably altered by Tie2 deletion. However, the veracity of these data is not clear given the questions about residual Tie2 expression, and this impacts the correlations in Fig 5C. Moreover, many of the statistically significant changes in gene expression appear trivial, such as the 12% reduction in Tie1 mRNA and the 17% increase in Ang2 expression in Tie2-deficient hearts (Fig 4A,C), thus it is not clear whether these changes have any biological significance. Without testing this in some physiological or pathophysiological model, as suggested in point #2, the relevance of these findings remains unclear.

Response to Rev2-5: In the manuscript we showed mRNA levels based on assessment with a primer/probe pair for the E7/E8 boundary. In our ‘Response to Rev2-1’ we demonstrated that these levels reflect full length residual Tie2. Therefore, the observations in the correlation plots in Fig 5C are considered relevant. 

We agree that the marginal increase in renal arteriolar Tie1 and Ang2 require further investigation to reveal whether this finding is biologically relevant, and calls for further studies. We have added the following to the Discussion section to highlight this ( line 402-407):

Similarly enigmatic is the observation that in glomeruli of transgenic mice, the balance in Tie2 /Tie1 expression, such as is present under control conditions, is altered. No such alterations were present in arterioles. Methods that enable the localization and quantitation of the four proteins of the Ang/Tie family studied here, and that can be combined with quantitation of Tie2 phosphorylation in discrete microvascular segments will be crucial for elucidating how these molecular systems contribute to heterogeneous microvascular endothelial behavior. 

Minor comment Reviewer 2: 

Rev2-6: Tie2 nomenclature (line 42) – The original name for Tie2 was Tunica interna Endothelial cell Kinase, or TEK (still the primary gene name) (Dumont, Oncogene, 1992). If the authors wish to acknowledge any name besides TIE2, it is suggested that this names be used, as “Tyrosine kinase receptor 2” is generic and not an accepted alias, although the related “Tyrosine-Protein Kinase Receptor TIE-2” is listed in GeneCards.

Response to Rev2-6: “Tyrosine kinase receptor 2” of the original manuscript was replaced by “Tyrosine-protein kinase receptor Tie2, also known as Tunica interna Endothelial cell Kinase or TEK” in the revised version of the manuscript at line 27

Reviewer #3:

Suggestion Rev3-1: From what I gather from the Methods, the animals were injected with TMX 3 times, once each in 3 successive weeks, at the dose listed, but at what age did the injections start? Whether this is an adult-age deletion or early-postnatal is critical to all of the results, but this information was not given. But this information has to be clearly stated in Methods. And equally importantly, since, the conclusions drawn depend on the age of deletion, the Discussion include comments on what this means in light of whether this was adult or early postnatal deletion (or other).

Response to Rev3-1: The interest of our research focuses on microvascular endothelial heterogeneity in adult tissues, on how intrinsic heterogeneity relates to differences in responses of microvascular segments to disease conditions, and on what the pharmacological consequences thereof are. 

For this reason, we worked with adult mice that were between 10-24 weeks of age when they received the first tamoxifen i.p. injection. We have stated this information in the Materials and Methods section, line 82/83 and 91/92.

---

## [Decision Letter · Decision Letter 1]

30 Sep 2021

PONE-D-21-04716R1Pattern of inducible endothelial Tie2 knockout in mature mouse microvasculature is organ- and vascular compartment-dependentPLOS ONE

Dear Dr. Zwiers,

Thank you for submitting your manuscript to PLOS ONE. After careful consideration, we feel that it has merit but does not fully meet PLOS ONE’s publication criteria as it currently stands. Therefore, we invite you to submit a revised version of the manuscript that addresses the points raised during the review process.

This revised manuscript still didn't satisfy reviewers comments.

Two of reviewers still have many issues with revised manuscript.

Rev. 1 suggests to reject it since "Authors did not properly address critical comments provided for previous version. Manuscript is still lacking methodological rigor, results are presented in poor manner and conclusions expressed in the discussion are not supported by findings."

Rev. 2, also having difficulty determining the main point of this paper.

Authors didn't provide any results to show efficient truncation, however come to conclusions that are not supported by data.

** This paper should be heavily revised prior to submission to another round of revision.**

Please include the following items when submitting your revised manuscript:A rebuttal letter that responds to each point raised by the academic editor and reviewer(s). You should upload this letter as a separate file labeled 'Response to Reviewers'.A marked-up copy of your manuscript that highlights changes made to the original version. You should upload this as a separate file labeled 'Revised Manuscript with Track Changes'.An unmarked version of your revised paper without tracked changes. You should upload this as a separate file labeled 'Manuscript'.

We look forward to receiving your revised manuscript.

Kind regards,

Slava Rom, Ph.D., MBA

Academic Editor

PLOS ONE

Reviewers' comments:

Reviewer's Responses to Questions

**Comments to the Author**

1. If the authors have adequately addressed your comments raised in a previous round of review and you feel that this manuscript is now acceptable for publication, you may indicate that here to bypass the “Comments to the Author” section, enter your conflict of interest statement in the “Confidential to Editor” section, and submit your "Accept" recommendation.

Reviewer #1: (No Response)

Reviewer #2: (No Response)

Reviewer #3: All comments have been addressed

2. Is the manuscript technically sound, and do the data support the conclusions?

Reviewer #1: No

Reviewer #2: Partly

Reviewer #3: Yes

3. Has the statistical analysis been performed appropriately and rigorously? 

Reviewer #1: No

Reviewer #2: Yes

Reviewer #3: Yes

4. Have the authors made all data underlying the findings in their manuscript fully available?

Reviewer #1: Yes

Reviewer #2: Yes

Reviewer #3: Yes

5. Is the manuscript presented in an intelligible fashion and written in standard English?

Reviewer #1: Yes

Reviewer #2: Yes

Reviewer #3: Yes

6. Review Comments to the Author

Reviewer #1: This is a resubmission of the revised manuscript “Pattern of inducible endothelial Tie2 knockout in mature mouse microvasculature is organ- and compartment-dependent”

Authors did not properly address critical comments provided for previous version. Manuscript is still lacking methodological rigor, results are presented in poor manner and conclusions expressed in the discussion are not supported by findings. Therefore, I cannot recommend this manuscript for publication.

Reviewer #2: I’m having difficulty determining the main point of this paper. I agree from the responses to my prior concerns about truncated RNA and protein that the model works, i.e., that Tie2 is targeted by the exon 9 deletion, but it is clear that the deletion is inefficient and that this is due to inefficient Cre-mediated excision of Tie2 exon 9 with the SCL-Cre driver. However, there still seems to be an argument by the authors that this could be a good model for pathophysiological studies. That may be true, but when the western blot in Fig R2 shows almost no loss of Tie2 expression in lung (albeit with no loading control shown), I have to question this conclusion, irrespective of higher apparent levels of deletion in microvascular beds. If I saw that western as evidence of the validity of the model, I would conclude that there was no deletion.

The authors state that the SCL-Cre driver was chosen because of its apparently high efficiency based on the prior study using a LacZ reporter: “The results presented in this paper lead us to conclude that a direct relation between Cre-based LacZ expression and Cre-based Tie2 knockout cannot be established.” That may well be true, but the best approach to establish the model’s efficiency would be to use simultaneous Cre-mediated reporter expression, as suggested in the first review. I recognize that this is likely beyond the scope of this report, but the paper’s conclusions should be modified appropriately. In the revised discussion, the authors have done a good job of discussing the limitations of various Cre drivers.

The authors state at the end of the abstract that, “future studies using similar knockout strategies should include a meticulous analysis of the knockout extent of the gene of interest, prior to studying its role in pathological conditions, so that proper conclusions can be drawn.” I agree with this statement, which seems obvious, as such an approach is a critical part of demonstrating the validity of any model.

To me, the simple conclusion from this paper is that this particular targeted mutation paired with this Cre driver is variably efficient in deleting Tie2. If claims beyond this regarding the efficacy of the model are to be made, it would be important to show simultaneous co-expression of a reporter, as suggested, or perhaps just use a different Cre driver (such as VE-cadherin-Cre). Accordingly, the title is misleading, as it suggests that the heterogeneity is somehow Tie2-dependent rather than being Cre-dependent. A better title would be simply, “Pattern of endothelial Tie2 deletion in an SCL-Cre-ERT;Tie2floxed mouse line”.

Reviewer #3: (No Response)

7. PLOS authors have the option to publish the peer review history of their article (what does this mean?). If published, this will include your full peer review and any attached files.

Reviewer #1: No

Reviewer #2: No

Reviewer #3: No

---

## [Author Response · Author response to Decision Letter 1]

9 Nov 2021

Rebuttal for PONE-D-21-04716: “Pattern of inducible endothelial Tie2 knockout in mature mouse microvasculature is organ- and vascular compartment-dependent” by Zwiers et al.

We would like to thank the reviewers for their time invested to enable us to further improve our manuscript, and the editor for giving us the opportunity to respond to the comments and questions raised. 

Reviewer #1.

Comment Rev1-1: Authors did not properly address critical comments provided for previous version. Manuscript is still lacking methodological rigor, results are presented in poor manner and conclusions expressed in the discussion are not supported by findings. 

Response to Rev1-1: In the first round of review, we extensively addressed the issues brought up by reviewer #1. Summarizing, we described choices regarding experimental design, including the lack of availability of antibodies to stain Tie1 protein in mouse tissues, amended the manuscript with requested additional statistical information, and improved textual parts of the manuscript. As the reviewer does not detail which issues were not properly addressed, it is not possible for us to (re)act. Furthermore, the criticism brought up by this reviewer is not shared by the other two reviewers. Instead, they assess the manuscript as technically sound, the data supporting conclusions (except for issues raised by reviewer #2 which we address below), the statistical analyses appropriate and rigorous, and the manuscript presented in an intelligible fashion. 

Reviewer #2.

Comments Rev2-1: I agree from the responses to my prior concerns about truncated RNA and protein that the model works, i.e., that Tie2 is targeted by the exon 9 deletion, but it is clear that the deletion is inefficient and that this is 

due to inefficient Cre-mediated excision of Tie2 exon 9 with the SCL-Cre driver.

Response to Rev2-1: We are happy to read that, based on the additional data generated and added to the revised version of the manuscript, the reviewer agrees that the conclusion is justified that the inducible, endothelial cell specific Tie2 knockout model works. 

Comment Rev2-2: However, there still seems to be an argument by the authors that this could be a good model for pathophysiological studies. That may be true, but when the western blot in Fig R2 shows almost no loss of Tie2 expression in lung (albeit with no loading control shown), I have to question this conclusion, irrespective of higher apparent levels of deletion in microvascular beds. If I saw that western as evidence of the validity of the model, I would conclude that there was no deletion.

Response to Rev2-2: We would like to stress that Figure R2 in our initial rebuttal aimed to answer this reviewer’s question whether truncated protein had been formed upon TMX treatment. Accordingly, the Western Blot analysis of Tie2 protein was designed to optimally detect intact Tie2 protein and potential truncation fragments thereof. From this data we concluded that no truncated protein was produced, which, in combination with PCR analyses using different primer sets, made us, and this reviewer, conclude that the mouse model works.

Figure 2 in the manuscript presents the quantitation of Tie2 protein by ELISA, which is the broadly accepted, superior technique for this purpose. The data in Figure 2 revealed loss of Tie2 protein in lungs, heart and aorta of TMX-treated animals. These quantitative data next led us to perform immunohistochemical staining of Tie2 in the organs, to identify which endothelial cells had lost the protein. We were as surprised as this reviewer to see the heterogeneous patterning of Tie2 loss shown in Fig 3A, and were even more surprised that this heterogeneous loss was highly consistent, as shown in figure 3B. 

In our search for published studies reporting TMX-inducible endothelial knockout mouse models to compare our data with, we discovered that in many papers, knockout was proven by exposing endothelial cells isolated from WT and TG animals, to OH-TMX in vitro, without providing further knockout evidence in vivo, or by analysis of target gene knockout in endothelial cells in one organ, while (patho)physiology studies focussed on another organ or other vascular segment [1–4]. Other studies used a reporter model to demonstrate the vascular beds in which the driver for Cre was activated by TMX, yet failed to report whether the target protein of interest was indeed knocked out in endothelial cells in these vascular beds [1,5]. 

As we consider quantitation of target gene knockout when using such models of importance for the reader, we briefly discuss our outcomes in the context of what others have published in the Discussion section:

Line 369: Combining Tie2floxed/floxed transgenic mice with end-SCL-Cre-ERT transgenic mice resulted in variable, yet highly reproducible Tie2 knockout, which was dependent on the location of the endothelial cells in the microvasculature in organs. Future studies will assess whether crossing of the Tie2floxed/floxed mouse with e.g., Cdh5CreERT2 or Flk-1-Cre mice [37, 38] yields knockout of Tie2 to the same extent, and in a similar pattern, as we report here. Knocking out another gene in the same Cre-driver mouse may result in different patterns of target gene loss [35,36,39]. Important to note is that many studies rely on reporter models to demonstrate where Cre recombinase is active, show target gene knockout in an organ which is not the organ under investigation for functional studies, or only show target gene knockout by in vitro exposure to tamoxifen. Information about the extent and location of target gene knockout is the cornerstone of this kind of studies to safeguard validity of the conclusions drawn, especially since extensive molecular heterogeneity exists between endothelial cells [40,41] which can have major functional consequences [42].

An elegant paper by Koh et al [6] shows in a Cdh5-Cre-ERT mouse model that 85% knockout of Tie2 in the brains of mice has occurred after short term treatment with TMX when these mice are 8 weeks old. They next study whether Tie2 has a role in a particular pathophysiological process in the brain. This study is an exception to the examples referred to above, as it demonstrates loss of target protein in the organ under study in a pathophysiological process. The mouse model Koh et al. use is the model reported by Savant et al., in which in the lungs of mice 6 days after birth, 25-33% Tie2 knockout had occurred using a similar TMX dosing regimen [4]. We added this information to the existing paragraph, discussing the previously reported study by Savant on endothelial knockout of Tie2 using Cre recombinase:

Line 398: An elegant study employing the same mouse model to investigate whether Tie2 exerts a role in tissue repair after photothrombotic injury evoked in the brain of 8 week old mice, showed 85% reduction of Tie2 protein expression in the brain [45]. It may well be that also in this Cdh5CreERT2 mouse model the extent of knockout of Tie2 depends on the organ. 

Comment Rev2-3: I’m having difficulty determining the main point of this paper.

Response to Rev2-3: The main point of this paper, is that knocking out Tie2 in endothelial cells in vivo using the SCL-Cre-ERT system leads to variation in extent of knockout in organs and microvascular segments in these organs. A corollary to this is that meticulous analysis of where the loss of target protein happens is crucial for this model, to prevent wrongful extrapolation from one organ to the other, or from one vessel to the other, a notion supported by this reviewer. We agree to the fullest with this reviewer that this ‘seems obvious, as such an approach is a critical part of demonstrating the validity of any model’, yet, as described in ‘Response to Rev2-2’, many studies, also using other drivers than SCL, steer away from demonstrating target gene knockout in the microvascular bed studied in (patho)physiological processes. We expect that the main point of our manuscript will create awareness regarding the importance of analysing the extent and location of knockout, and have added a paragraph on this matter to the Discussion section (line 392-401).

Comment Rev2-4: The authors state that the SCL-Cre driver was chosen because of its apparently high efficiency based on the prior study using a LacZ reporter: “The results presented in this paper lead us to conclude that a direct relation between Cre-based LacZ expression and Cre-based Tie2 knockout cannot be established.” That may well be true, but the best approach to establish the model’s efficiency would be to use simultaneous Cre-mediated reporter expression, as suggested in the first review. I recognize that this is likely beyond the scope of this report […].

Response to Rev2-4: Investigating mechanisms to explain the heterogeneous patterning of Tie2 loss was, and is, indeed, beyond the scope of our study. 

Yet, based on this reviewer’s comment, we tried to get a glimpse of the location of activated, i.e., nuclear Cre recombinase, and to relate this to Tie2 expression. For this purpose, we fluorescently double-stained kidney sections of C57BL/6 wildtype mice, TMX-treated Tie2fl/fl/Cre- control mice, and TMX treated Tie2�E9 knockout mice, with anti-Cre and anti-Tie2 antibodies. We used two different anti-Cre-recombinase antibodies, i.c., Rabbit polyclonal Ab40011 (Abcam, Cambridge, UK) and Rabbit polyclonal cat#908001 (BioLegend, San Diego, USA). Unfortunately, both antibodies showed significant non-specific binding in all three groups of mice, which we were not able to block (data not shown). We could therefore not draw any conclusions on the exact location of Cre-recombinase and Tie2 deletion in Tie2�E9 knockout mice, to explain, at least in part, the observed heterogeneous pattern of Tie2 knockout. 

Comment Rev2-5: In the revised discussion, the authors have done a good job of discussing the limitations of various Cre drivers. […] To me, the simple conclusion from this paper is that this particular targeted mutation paired with this Cre driver is variably efficient in deleting Tie2. If claims beyond this regarding the efficacy of the model are to be made, it would be important to show simultaneous co-expression of a reporter, as suggested, or perhaps just use a different Cre driver (such as VE-cadherin-Cre).

Response to Rev2-5: We agree with this reviewer that we cannot and should not extrapolate our findings to any other mouse model, and therefore have carefully rephrased generalized statements into dedicated information (highlighted yellow in the revised manuscript) about the model which we here report:

Line 319: For this purpose, we used the end-SCL-Cre-ERT mouse model, in which tamoxifen-inducible Cre ERT recombinase expression was driven by the endothelial enhancer of the Stem Cell Leukemia locus. In this mouse model, using LacZ as reporter, Cre recombinase was shown to be active in the microvasculature of several, though not all, organs [24].

We furthermore heavily revised the section on using other driver-Cre mouse models, as described in our ‘Response to Rev2-2’ (line 369-380 and 392-401).

Comment Rev2-6: Accordingly, the title is misleading, as it suggests that the heterogeneity is somehow Tie2-dependent rather than being Cre-dependent. A better title would be simply, “Pattern of endothelial Tie2 deletion in an SCL-Cre-ERT;Tie2floxed mouse line”.

Response to Rev2-6: We have adapted the title as advised by the reviewer: Pattern of tamoxifen-induced Tie2 deletion in endothelial cells in mature blood vessels using endo SCL-Cre-ERT transgenic mice.

Reviewer #3.

Reviewer #3 judges the manuscript technically sound, with data supporting the conclusions, with statistical analyses performed appropriately and rigorously, and presented in an intelligible fashion and written in standard English and has no further comments.

References

1. Korhonen H, Fisslthaler B, Moers A, Wirth A, Habermehl D, Wieland T, et al. Anaphylactic shock depends on endothelial Gq/G11. J Exp Med. 2009;206: 411–420. doi:10.1084/jem.20082150

2. Langer HF, Orlova V V., Xie C, Kaul S, Schneider D, Lonsdorf AS, et al. A Novel Function of Junctional Adhesion Molecule-C in Mediating Melanoma Cell Metastasis. Cancer Res. 2011;71: 4096–4105. doi:10.1158/0008-5472.CAN-10-2794

3. Schmidt TT, Tauseef M, Yue L, Bonini MG, Gothert J, Shen T-L, et al. Conditional deletion of FAK in mice endothelium disrupts lung vascular barrier function due to destabilization of RhoA and Rac1 activities. Am J Physiol Cell Mol Physiol. 2013;305: L291–L300. doi:10.1152/ajplung.00094.2013

4. Savant S, La Porta S, Budnik A, Busch K, Hu J, Tisch N, et al. The Orphan Receptor Tie1 Controls Angiogenesis and Vascular Remodeling by Differentially Regulating Tie2 in Tip and Stalk Cells. Cell Rep. 2015;12: 1761–1773. doi:10.1016/j.celrep.2015.08.024

5. Perry HM, Huang L, Ye H, Liu C, Sung SJ, Lynch KR, et al. Endothelial Sphingosine 1‑Phosphate Receptor‑1 Mediates Protection and Recovery from Acute Kidney Injury. J Am Soc Nephrol. 2016;27: 3383–3393. doi:10.1681/ASN.2015080922

6. Koh BI, Lee HJ, Kwak PA, Yang MJ, Kim J-H, Kim H-S, et al. VEGFR2 signaling drives meningeal vascular regeneration upon head injury. Nat Commun. 2020;11: 3866. doi:10.1038/s41467-020-17545-2

---

## [Decision Letter · Decision Letter 2]

14 Feb 2022

PONE-D-21-04716R2Pattern of tamoxifen-induced Tie2 deletion in endothelial cells in mature blood vessels using endo SCL-Cre-ERT transgenic micePLOS ONE

Dear Dr. Zwiers,

Thank you for submitting your manuscript to PLOS ONE. After careful consideration, we feel that it has merit but does not fully meet PLOS ONE’s publication criteria as it currently stands. Therefore, we invite you to submit a revised version of the manuscript that addresses the points raised during the review process.

Please address the requirements from Staff Editors set in '**Journal Requirements**' section, such that this manuscript is amended to meet PLOS ONE's publication criteria.

We look forward to receiving your revised manuscript.

Kind regards,

Slava Rom, Ph.D.

Academic Editor

PLOS ONE

**Journal Requirements:**

Please address the comments from reviewer 1 regarding reporting of statistics/data availability. 

Please note that as this manuscript stands it does not meet PLOS ONE's publication criteria for data availability and reporting of statistics. In the reporting of statistical results (https://journals.plos.org/plosone/s/submission-guidelines#loc-statistical-reporting) in the section ‘Reporting of statistical results’ our policies state that ‘Results must be rigorously and appropriately reported, in keeping with community standards’ such that for:

**Properties of distribution**. It should be clear from the text which measures of variance (standard deviation, standard error of the mean, confidence intervals) and central tendency (mean, median) are being presented.**Regression analyses.** Include the full results of any regression analysis performed as a supplementary file. Include all estimated regression coefficients, their standard error, p-values, and confidence intervals, as well as the measures of goodness of fit.**Reporting parameters**. Test statistics (F/t/r) and associated degrees of freedom should be provided. Effect sizes and confidence intervals should be reported where appropriate. If percentages are provided, the numerator and denominator should also be given.**Open data.** As explained in PLOS’s Data Policy (https://journals.plos.org/plosone/s/data-availability), be sure to make individual data points, underlying graphs and summary statistics available at the time of publication. Data can be deposited in a repository or included within the Supporting Information files.

Please update the results section to ensure that these reporting guidelines are met, and as the manuscript states in the data availability statement that 'all relevant data are within the manuscript and supporting information files' please upload the data that underlies the figures/summary statistics in this manuscript as supporting information, or upload it to a repository, and update the data availability statement with a link to the appropriate repository, for example 'All underlying data files are available from the XXX database (accession number(s) XXX, XXX.)'

Additional Editor Comments (if provided):

Based on 2 reviewers suggestions, this manuscript is ready to be accepted.

Reviewers' comments:

Reviewer's Responses to Questions

**Comments to the Author**

1. If the authors have adequately addressed your comments raised in a previous round of review and you feel that this manuscript is now acceptable for publication, you may indicate that here to bypass the “Comments to the Author” section, enter your conflict of interest statement in the “Confidential to Editor” section, and submit your "Accept" recommendation.

Reviewer #1: (No Response)

Reviewer #3: All comments have been addressed

Reviewer #4: All comments have been addressed

2. Is the manuscript technically sound, and do the data support the conclusions?

Reviewer #1: Partly

Reviewer #3: (No Response)

Reviewer #4: Yes

3. Has the statistical analysis been performed appropriately and rigorously? 

Reviewer #1: No

Reviewer #3: Yes

Reviewer #4: Yes

4. Have the authors made all data underlying the findings in their manuscript fully available?

Reviewer #1: No

Reviewer #3: Yes

Reviewer #4: Yes

5. Is the manuscript presented in an intelligible fashion and written in standard English?

Reviewer #1: Yes

Reviewer #3: Yes

Reviewer #4: Yes

6. Review Comments to the Author

Reviewer #1: In the first revision (and in the 2nd revision as well) authors failed to address first set of comments provided for the initial submission. As a result this manuscript still lacks important information such as:

1. Results section is still incomplete. Authors state that data is presented as mean and SD, however no actual values are included in Results section or Figure legends.

2. Still no scale bar in the IHC panels (Fig 3).

3. In Fig 2B Tie2 quantitation by ELISA is presented. However in the Methods section there is no information about reagents (kits) used in this experiment and about sensitivity range of the assay, which is important given that authors made conclusions based on quite low levels of protein (within 0-1 pg/ml range, unfortunately it is impossible to determine exact values from the graph bar and authors have not provided that information in Results section or Figure legend).

Reviewer #3: I have no further concerns. The authors have discussed the limitations of their study. With these changes, we are set.

Reviewer #4: The authors have submitted a revised manuscript which is improved. I have no concerns regarding the manuscript.

7. PLOS authors have the option to publish the peer review history of their article (what does this mean?). If published, this will include your full peer review and any attached files.

Reviewer #1: No

Reviewer #3: No

Reviewer #4: No

---

## [Author Response · Author response to Decision Letter 2]

17 Mar 2022

Rebuttal for PONE-D-21-04716: “Pattern of tamoxifen-induced Tie2 deletion in endothelial cells in mature blood vessels using endo SCL-Cre-ERT transgenic mice” by Zwiers et al.

We would like to thank the reviewers for their time invested to enable us to further improve our manuscript, and the editor for giving us the opportunity to respond to the comments and questions raised. 

Journal Requirements:

1-1: Please address the comments from reviewer 1 regarding reporting of statistics/data availability.

Response to journal requirements: To fulfill the journal requirements and addressing the comments by reviewer 1 regarding reporting of statistics and data availability we have uploaded all GraphPad files as supplementary data as well as the raw Excel data files that belong to the ELISA read-out. We have amended the manuscript text as well as the supplementary data titles accordingly.

Line 128: Raw data files are available in supplementary data files (S9 Dataset).

Line 160: All GraphPad PZFX-files are available as supplementary data (S10-16 Dataset).

Line 659-676:

S9 Dataset. Excel data file that represents all raw data points of the Tie2 ELISA used for Fig 2B. Standard curve is shown for samples measured in one run.

S10 Dataset. GraphPad data file used for the statistical analysis of mRNA expression levels as presented in Fig 2A, Fig 4A-C, and S4 Table.

S11 Dataset. GraphPad data file used for the statistical analysis of Tie2 protein levels (ELISA) as presented in Fig 2B.

S12 Dataset. GraphPad data file used for the statistical analysis of Tie2 protein localization in different organs (IHC) by morphometrics as presented in Fig 3B.

S13 Dataset. GraphPad data file used for the statistical analysis of mRNA expression levels in laser microdissected compartments as presented in Fig 5B-E.

S14 Dataset. GraphPad data file used for the statistical analysis of Tie1-Tie2 correlations in laser microdissected compartments as presented in Fig 5F,G.

S15 Dataset. GraphPad data file used for the statistical analysis of VE-cadherin and CD31 mRNA expression levels as presented in S3 Fig.

S16 Dataset. GraphPad data file used for the statistical analysis of Tie2/Tie1 and Ang1/Ang2 mRNA level ratios as presented in S6 and S7 Tables.

S17 Dataset. GraphPad data file used for the statistical analysis of Tie2 mRNA expression levels using 3 different primer/probe sets as presented in S5 Fig.

1-2: Please review your reference list to ensure that it is complete and correct. If you have cited papers that have been retracted, please include the rationale for doing so in the manuscript text, or remove these references and replace them with relevant current references. Any changes to the reference list should be mentioned in the rebuttal letter that accompanies your revised manuscript. If you need to cite a retracted article, indicate the article’s retracted status in the References list and also include a citation and full reference for the retraction notice.

Response to review reference list: We have checked all the references we have used in this manuscript. The list is complete, none of the cited papers have been retracted, hence we did not make any changes.

Reviewer #1.

Comment Rev1-1: Results section is still incomplete. Authors state that data is presented as mean and SD, however no actual values are included in Results section or Figure legends. 

Response to Rev1-1: We indeed did not present the data as mean values explicitly in the paper as we judged that this might compromise the readability of the paper. SD however is shown in each figure by error bars and mentioned as such in figure legends. Since reviewer #1 judges loss of readability not an issue, we have included the mean, SD, and p values and we adapted the text in the Results section according to her/his advice. Furthermore we have added the following to all relevant figure legends (changes in highlights): ”Graphs show individual values and means (black lines) ± SD (error bars)”.

Line 205-214: In Tie2�E9 knockout mice, Tie2 mRNA expression was significantly reduced in kidney (25%; mean Tie2�E9= 0.59 (SD 0.08); mean Tie2fl/fl/Cre-= 0.79 (SD 0.12); p= 0.0026), lung (55%; mean Tie2�E9= 0.12 (SD 0.05); mean Tie2fl/fl/Cre-= 0.27 (SD 0.06); p= 0.0002), heart (35%; mean Tie2�E9= 0.15 (SD 0.03); mean Tie2fl/fl/Cre-= 0.23 (SD 0.03); p<0.0001), and aorta (38%; mean Tie2�E9= 0.13 (SD 0.07); mean Tie2fl/fl/Cre-= 0.21 (SD 0.04); p= 0.013), but not in liver (mean Tie2�E9= 0.17 (SD 0.04); mean Tie2fl/fl/Cre-= 0.18 (SD 0.05); p= 0.54) (Fig 2A). This organ-specific reduction of Tie2 mRNA was paralleled by a reduction in Tie2 protein in lung (66%; mean Tie2�E9= 2.45 (SD 1.10); mean Tie2fl/fl/Cre-= 7.21 (SD 3.01); p= 0.0017), heart (37%; mean Tie2�E9= 0.31 (SD 0.05); mean Tie2fl/fl/Cre-= 0.50 (SD 0.06); p<0.0001), and aorta (51%; mean Tie2�E9= 0.36 (SD 0.18); mean Tie2fl/fl/Cre-= 0.73 (SD 0.22); p= 0.0033), as assessed by ELISA on whole organ homogenates (Fig 2B).

Line 241-254: When Tie2 protein staining was quantified by morphometry in whole organ sections of Tie2�E9 knockout and Tie2fl/fl/Cre- control mice, a statistically significant Tie2 protein reduction was identified in lung (79%; mean Tie2�E9= 3.50 (SD 2.83); mean Tie2fl/fl/Cre-= 16.52 (SD 5.96); p= 0.0002) and aorta (57%; mean Tie2�E9= 2.34 (SD 0.98); mean Tie2fl/fl/Cre-= 5.45 (SD 1.83); p= 0.0015) (Fig 3B, ‘total’).

We next refined quantification of Tie2 protein by selectively focusing on microvascular compartments (Fig 3B). This approach uncovered a significant reduction in Tie2 protein expression in arterioles (70%; mean Tie2�E9= 8.27 (SD 4.94); mean Tie2fl/fl/Cre-= 27.81 (SD 7.46); p<0.0001) and glomeruli (17%; mean Tie2�E9= 35.84 (SD 5.36); mean Tie2fl/fl/Cre-= 43.04 (SD 4.75); p= 0.0163) of the kidney, which had been concealed by whole-tissue quantification. Moreover, a significant reduction in Tie2 protein was detected in arterioles of the heart (83%; mean Tie2�E9= 1.86 (SD 1.08); mean Tie2fl/fl/Cre-= 10.91 (SD 8.93); p= 0.02), which, similarly, had been obscured by whole-tissue quantification. Vascular compartment-specific quantification in lung revealed a significant reduction of Tie2 protein in both arterioles (70%; mean Tie2�E9= 13.51 (SD 5.41); mean Tie2fl/fl/Cre-= 45.24 (SD 8.05); p<0.0001) and capillaries (85%; mean Tie2�E9= 3.01 (SD 4.22); mean Tie2fl/fl/Cre-= 20.18 (SD 7.45); p= 0.0001) (Fig 3B). 

Line 277-287: In Tie2�E9 knockout mice, Tie1 mRNA expression was significantly reduced in kidney (19%; mean Tie2�E9= 0.40 (SD 0.07); mean Tie2fl/fl/Cre-= 0.49 (SD 0.08); p= 0.0303), lung (23%; mean Tie2�E9= 0.16 (SD 0.01); mean Tie2fl/fl/Cre-= 0.21 (SD 0.03); p= 0.0012) and heart (12%; mean Tie2�E9= 0.35 (SD 0.04); mean Tie2fl/fl/Cre-= 0.40 (SD 0.03); p= 0.0155) (Fig 4A), i.e. in organs that also exhibited a significant reduction in Tie2 mRNA levels. In contrast, the significant Tie2 mRNA reduction in aorta of Tie2�E9 knockout mice was not accompanied by a reduction in Tie1 mRNA expression. Tie1 mRNA expression in liver did not differ between genotypes. 

Endothelial Tie2�E9knockout did not affect Ang1- or Ang2 mRNA expression in most organs (Figs 4B and C), with the exception of the heart, where Ang2 mRNA expression was significantly increased (17%; mean Tie2�E9= 0.0066 (SD 0.0007); mean Tie2fl/fl/Cre-= 0.0056 (SD 0.0008); p= 0.0252) in Tie2�E9 knockout mice (Fig 4C). 

Line 310-323: Deletion of Tie2 exon 9 was accompanied by significantly reduced Tie2 mRNA levels in glomeruli (mean Tie2�E9= 0.80 (SD 0.14); mean Tie2fl/fl/Cre-= 1.01 (SD 0.13); p= 0.009). In arterioles from control mice high variation in Tie2 expression was observed, with no statistically significant differences between the two groups (mean Tie2�E9= 0.67 (SD 0.19); mean Tie2fl/fl/Cre-= 0.87 (SD 0.36); p= 0.21) (Fig 5B), contrasting significantly lower Tie2 protein expression in the Tie2�E9 knockout mice. Tie1 mRNA expression was significantly increased in arterioles (mean Tie2�E9= 1.38 (SD 0.26); mean Tie2fl/fl/Cre-= 0.86 (SD 0.35); p= 0.007), but not in glomeruli (mean Tie2�E9= 0.40 (SD 0.03); mean Tie2fl/fl/Cre-= 0.47 (SD 0.08); p= 0.06) of Tie2�E9 knockout mice (Fig 5C). Analysis of gene expression in kidney microvascular compartments also revealed a significant, two-fold increase in Ang1 expression in arterioles of Tie2�E9 knockout mice (mean Tie2�E9= 0.107 (SD 0.045); mean Tie2fl/fl/Cre-= 0.059 (SD 0.035); p= 0.036), but not in glomeruli (mean Tie2�E9= 0.022 (SD 0.007); mean Tie2fl/fl/Cre-= 0.023 (SD 0.008); p= 0.70) (Fig 5D), while Ang2 expression was unchanged in arterioles (mean Tie2�E9= 0.016 (SD 0.012); mean Tie2fl/fl/Cre-= 0.017 (SD 0.011); p= 0.80) and glomeruli (mean Tie2�E9= 0.054 (SD 0.004), mean Tie2fl/fl/Cre-= 0.056 (SD 0.016); p= 0.78) (Fig 5E).

S4 table and S7 table: we replaced NS by the p values.

Comment Rev1-2: Still no scale bar in the IHC panels (Fig 3).

Response to Rev1-2: We indeed did not add a scalebar in the IHC panels. Instead, we explicitly mentioned the magnification of each photomicrograph in figure legend 3A. We decided to take this approach as we were not able to add scale bars in a legible manner due to the large number and small size of the photomicrographs. 

Based on this comment by the reviewer, we considered adding a scale bar to each photomicrograph, but by doing so we would falsify this figure. From our perspective, our initially chosen approach provides readers sufficient information to judge our results and to perform similar analyses in their own laboratory with our M&M and results as a reference. To allow readers to dive further into the details of our results, we propose to upload our Hamamatsu scans (NPDI-files, total size= 33GB) to a PLOS ONE repository, for everybody to access, provided PLOS ONE has such a repository available. A third party solution could be the freely available Zenodo database, provided by the CERN Data Platform to support open access data availability, that uniquely tags the scans by a DOI. We ask the editor for advice on this, and will adapt the manuscript accordingly. We will provide a short manual how to access the scans by use of Aperio ImageScope software.

We propose to add a sentence in the Methods sections to point to the DOI for NPDI-file access:

Line 144: Raw NPDI files are available via repository DOIxxx.

Comment Rev1-3: In Fig 2B Tie2 quantitation by ELISA is presented. However in the Methods section there is no information about reagents (kits) used in this experiment and about sensitivity range of the assay, which is important given that authors made conclusions based on quite low levels of protein (within 0-1 pg/ml range, unfortunately it is impossible to determine exact values from the graph bar and authors have not provided that information in Results section or Figure legend).

Response to Rev1-3: We agree with reviewer #1 that the information on the ELISA was concise. To determine Tie2 protein in tissue homogenates, an input of 100 – 500 ug total protein in the tissue homogenate was chosen. With this input, an OD value was obtained which was approximately in the middle of the calibration curve (125 – 8,000 pg/mL). OD values of the samples were calculated to Tie2 protein using the calibration curve (Tie2 pg/ml), corrected for volume (Tie2 pg) and normalized to protein input (Tie2 pg/ug).

The text has been adapted in the Methods section:

Line 121-127: Briefly, cryosections of organs were used to prepare tissue homogenates in RIPA buffer containing protease inhibitor, phosphatase inhibitor and activated Na3VO4. Total protein concentration was determined by DC™ Protein Assay Kit II (#5000112; Bio-Rad Laboratories B.V., Veenendaal, The Netherlands). Tie2 protein expression was quantified by use of Mouse Tie2 Quantikine ELISA Kit (cat# MTE200, sensitivity 40.2 pg/ml and assay range 125.0 – 8,000 pg/ml) according to manufacturer’s instruction (R&D Systems, Minneapolis, MN, USA). Amounts of Tie2 protein were normalized for the total protein input of tissue homogenate and expressed as pg/µg of total protein.

---

## [Decision Letter · Decision Letter 3]

13 May 2022

Pattern of tamoxifen-induced Tie2 deletion in endothelial cells in mature blood vessels using endo SCL-Cre-ERT transgenic mice

PONE-D-21-04716R3

Dear Dr. Zwiers,

We’re pleased to inform you that your manuscript has been judged scientifically suitable for publication and will be formally accepted for publication once it meets all outstanding technical requirements.

Kind regards,

Slava Rom, Ph.D.

Academic Editor

PLOS ONE

Reviewers' comments:

Reviewer's Responses to Questions

**Comments to the Author**

1. If the authors have adequately addressed your comments raised in a previous round of review and you feel that this manuscript is now acceptable for publication, you may indicate that here to bypass the “Comments to the Author” section, enter your conflict of interest statement in the “Confidential to Editor” section, and submit your "Accept" recommendation.

Reviewer #1: All comments have been addressed

Reviewer #2: All comments have been addressed

Reviewer #5: (No Response)

2. Is the manuscript technically sound, and do the data support the conclusions?

Reviewer #1: Yes

Reviewer #2: Yes

Reviewer #5: Yes

3. Has the statistical analysis been performed appropriately and rigorously? 

Reviewer #1: Yes

Reviewer #2: Yes

Reviewer #5: Yes

4. Have the authors made all data underlying the findings in their manuscript fully available?

Reviewer #1: Yes

Reviewer #2: Yes

Reviewer #5: Yes

5. Is the manuscript presented in an intelligible fashion and written in standard English?

Reviewer #1: Yes

Reviewer #2: Yes

Reviewer #5: Yes

6. Review Comments to the Author

Reviewer #1: (No Response)

Reviewer #2: The authors have appropriately acknowledged the limitations of the SCL-Cre model, which is most likely to account for the variability in Tie2 deletion in various tissues.

Reviewer #5: I am assessing this manuscript for the first time and will restrict my review to the originality, soundness of the data and conclusions of the manuscript in its current revision. This is a careful study that reports novel data. The detailed examination of Tie2 expression in the different vascular beds within tissues is valuable. Particularly as it highlights the importance of assessing the extent of knockdown in the relevant tissues and vessel beds when attempting to correlate loss of Tie2 with tissue phenotype. The model reported will also be useful in future studies. The methodology is sound . Overall, the conclusions drawn by the authors are supported by the data, the analyses are appropriate and the data is reported in sufficient detail.

7. PLOS authors have the option to publish the peer review history of their article (what does this mean?). If published, this will include your full peer review and any attached files.

Reviewer #1: No

Reviewer #2: No

Reviewer #5: No

---

## [Editor Report · Acceptance letter]

23 May 2022

PONE-D-21-04716R3 

Pattern of tamoxifen-induced Tie2 deletion in endothelial cells in mature blood vessels using endo SCL-Cre-ERT transgenic mice 

Dear Dr. Zwiers:

I'm pleased to inform you that your manuscript has been deemed suitable for publication in PLOS ONE. Congratulations! Your manuscript is now with our production department. 

Kind regards, 

on behalf of

Dr. Slava Rom 

Academic Editor

PLOS ONE